# Both Nodal signalling and stochasticity select for prospective distal visceral endoderm in mouse embryos

Katsuyoshi Takaoka[1,2,3], Hiromi Nishimura[1,2] & Hiroshi Hamada[1,2]

Anterior–posterior (A–P) polarity of mouse embryos is established by distal visceral endoderm (DVE) at embryonic day (E) 5.5. *Lefty1* is expressed first at E3.5 in a subset of epiblast progenitor cells (L1$^{epi}$ cells) and then in a subset of primitive endoderm cells (L1$^{dve}$ cells) fated to become DVE. Here we studied how prospective DVE cells are selected. *Lefty1* expression in L1$^{epi}$ and L1$^{dve}$ cells depends on Nodal signaling. A cell that experiences the highest level of Nodal signaling begins to express *Lefty1* and becomes an L1$^{epi}$ cell. Deletion of *Lefty1* alone or together with *Lefty2* increased the number of prospective DVE cells. Ablation of L1$^{epi}$ or L1$^{dve}$ cells triggered *Lefty1* expression in a subset of remaining cells. Our results suggest that selection of prospective DVE cells is both random and regulated, and that a fixed prepattern for the A–P axis does not exist before the blastocyst stage.

---

[1] Developmental Genetics Group, Graduate School of Frontier Biosciences, Osaka University, 1-3 Yamada-oka, Suita, Osaka 565-0871, Japan. [2] RIKEN Center for Developmental Biology, 2-2-3 Minatojima-minamimachi, Chuo-ku, Kobe, Hyogo 650-0047, Japan. [3] Present address: Max Planck Institute for Biophysical Chemistry, Am Fassberg 11, 37077 Gottingen, Germany. Correspondence and requests for materials should be addressed to K.T. (email: katsuyoshi.takaoka@mpibpc.mpg.de) or to H.H. (email: hiroshi.hamada@riken.jp)

n *Drosophila*, the anterior–posterior (A–P) body axis is specified by maternal determinants that are asymmetrically distributed within the oocyte with respect to future A–P polarity[1]. Such maternal determinants do not appear to exist for mammals such as the mouse, however, with the mechanism by which A–P polarity is established in these animals having remained unknown. A–P polarity is established in the mouse embryo when the distal visceral endoderm (DVE) migrates toward the future anterior side at embryonic day (E) 5.5 (refs. [2–7]). Concomitant with DVE migration, all visceral endoderm (VE) cells in the

embryonic region undergo global movement, resulting in the localization of some VE cells at the distal tip of the embryo. These VE cells at the distal tip will become the anterior visceral endoderm (AVE) and migrate toward the future anterior side of the embryo by following the migration of DVE[8]. Development of the A–P axis is thus a self-organizing process that does not require maternal cues.[9,10]

Whereas A–P polarity of the mouse embryo is firmly established during the period from E5.5 to E6.5, its origin can be traced back to preimplantation stages of development. *Lefty1* is a marker of both DVE and AVE, but its expression begins in the blastocyst. It is expressed first in a subset of epiblast progenitor cells and then in a subset of primitive endoderm (PrE) progenitors, the latter of which is fated to become DVE. Expression of *Lefty1* therefore marks prospective DVE cells in peri-implantation embryos[8]. Although generation of Lefty1+ future DVE cells[9] and Cerl1+ DVE cells[10,11] occurs in an embryo-autonomous manner, generation of fully functional DVE may require interaction with the uterus[12]. Whereas Nodal signaling[13] and expression of its target gene *Eomes*[14] are essential for DVE formation, it has remained unknown how *Lefty1* expression is induced and how prospective DVE cells are selected in peri-implantation embryos. In this study, we have now addressed these questions by studying the regulation of *Lefty1* expression and its role in specification of future DVE cells. Our results suggest that selection of prospective DVE cells in mouse peri-implantation embryo is both random and regulated.

## Results

***Lefty1* expression is regulated by Nodal signaling.** We have previously shown that *Lefty1* is expressed first (at E3.5) in a subset of epiblast progenitor cells and then (between E3.75 and E4.5) in a subset of PrE progenitors fated to become DVE[8], with these Lefty1+ cell subsets being herein designated L1$^{epi}$ cells and L1$^{dve}$ cells, respectively. Some DVE cells were previously reported to be derived from epiblast (Sox2+ cells) that transmigrates into VE[12]. We examined this possibility by testing whether Oct3/4+ and Sox2+ epiblast contributes to DVE. We were unable to detect Oct3/4 (mTomato)+ cells (7/7 embryos at E5.5), Oct3/4+ cells (14/14 embryos at E5.5) or Sox2+ cells (4/4 embryos at E5.5, 5/5 embryos at E6.0) in the DVE region (Supplementary Fig. 1), however, suggesting that all DVE cells are derived from L1$^{dve}$ cells between E3.75 and E4.5, as we previously described[8].

We examined how *Lefty1* expression is regulated in both L1$^{epi}$ and L1$^{dve}$ cells (Fig. 1). A *Lefty1(mVenus* or *Cherry)* bacterial artificial chromosome (BAC) transgene that recapitulates *Lefty1* expression in embryos[8] was active in epiblast progenitor cells[8] within the inner cell mass (ICM) of E3.5 embryos and in the PrE of E4.5 embryos[8,9] (Supplementary Fig. 2a, b, c), representing *Lefty1* expression in L1$^{epi}$ and L1$^{dve}$ cells, respectively. *L1-10.5-Venus*, a transgene that contains the 10.5-kb upstream region of *Lefty1* and which recapitulates *Lefty1* expression at E6.5 and E8.0 (refs. [9,15]) (Fig. 1b), was also active at E3.5 (presumably in L1$^{epi}$ cells) and at E4.5 (presumably in L1$^{dve}$ cells) (Fig. 1b).

Given that left–right (L–R) asymmetric expression of *Lefty1* at E8.0 is regulated by Nodal-Foxh1 signaling[15], we examined the possible role of such signaling in *Lefty1* expression at E3.5 and E4.5. Culture of E3.2 embryos harboring a *Lefty1(mVenus)* BAC transgene with the Nodal signaling inhibitor SB431542 for 24 h prevented the emergence of *Lefty1* expression (11/11 embryos) (Supplementary Fig. 2h). Foxh1-binding sequences that are conserved between mouse and human[9] are present within the 10.5-kb upstream region of *Lefty1*, with two such sequences being located in the region around −1.5 kb and two in the region around −10 kb (ref. [9]). Foxh1 binding has been detected at both of these regions in human embryonic stem cells by chromatin immunoprecipitation–sequencing (ChIP-seq) analysis[16,17] (Supplementary Fig. 2i). The Foxh1-dependent enhancers located at −10 and −1.5 kb of *Lefty1* are hereafter referred to as DE (distal enhancer) and PE (proximal enhancer), respectively.

The *L1-DE+PE+-mVenus* transgene, which contains DE and PE, was active in a few epiblast progenitor cells at E3.5 (L1$^{epi}$ cells) and in GATA6+ cells in PrE at E4.5 (L1$^{dve}$ cells) (Fig. 1a, b, and Supplementary Fig. 2b). *L1-PE+-Venus*, or *L1-PE+-mVenus* which contains PE but lacks DE, was inactive in the ICM at E3.5 but active in a few GATA6+ cells of PrE at E4.5 (L1$^{dve}$ cells) (Fig. 1a, b, and Supplementary Fig. 2c). Similarly, *L1-PE+-lacZ* was inactive at E3.5, active at E4.5, and inactive at E6.5 (Supplementary Fig. 2e). *L1-DE+ transgenes* were active in the ICM at E3.5 but showed ectopic expression in the epiblast between E4.5 and E6.5 (Fig. 1b and Supplementary Fig. 2f, g). *L1-DE+PE+-lacZ* was not expressed at E3.5, E4.5, or E6.5 in *Nodal−/−* embryos (11/12 embryos at E3.5, 8/8 embryos at E4.5, 2/2 embryos at E6.5) (Fig. 1c), suggesting that the activity of both DE and PE is Nodal dependent. Consistent with this notion, *L1-DE$^{m}$PE+-Cherry* or *L1-DE$^{m}$PE+-lacZ*, in which the Foxh1-binding sites of DE are mutated, was inactive at E3.5 but was active at E4.5 (Fig. 1a, b, and Supplementary Fig. 2d), whereas *L1-PE$^{m}$-lacZ*, in which the Foxh1-binding sequences of PE are mutated, was not active at E4.5 (Fig. 1b). Furthermore, SB431542 abolished expression of *L1-DE+PE+-mVenus* (6/6 embryos) and *L1-PE+-Venus* (4/4 embryos) in E3.2 embryos cultured for 24 h (Supplementary Fig. 2h).

*L1-DE$^{m}$PE+-Cre*, a *Cre* transgene driven by PE, marked DVE at E5.5 and DVE-derived cells at E6.5 but failed to label epiblast at both stages (24/24 embryos) (Fig. 1d). Similarly, *L1-2.2-Cre*, which contains PE, specifically marked DVE-derived cells, excluding the epiblast, at E6.5 (6/7 embryos) (Fig. 1d). Together, these results suggested that the Foxh1-binding sites in DE are essential for *Lefty1* expression in L1$^{epi}$ cells of the ICM at E3.5,

**Fig. 1** *Lefty1* expression in L1$^{epi}$ and L1$^{dve}$ cells is regulated by Nodal-Foxh1 signaling. **a** Expression of three *Lefty1* transgenes (*L1-DE+PE+-mVenus*, *L1-DE$^{m}$PE+-Cherry*, and *L1-PE+-Venus*) was examined in mouse embryos at E3.5 and E4.5. Embryos were immunostained for transgene expression as well as for GATA6 (a PrE-specific marker), and they were counterstained with 4',6-diamidino-2-phenylindole (DAPI). Bright-field images are also shown. The expression pattern of *L1-DE+PE+-lacZ* in wild-type embryos has been described previously[8]. The number of cells in each embryo is indicated. Scale bars, 50 μm. **b** Structures of various *Lefty1* reporter transgenes and summary of their activities at the indicated stages. *L1-BAC* is the *Lefty1(mVenus)* BAC transgene generated by replacement of *lacZ* in the *Lefty1(lacZ)* BAC transgene[9] with *mVenus*. The positions of two Foxh1-dependent enhancers, DE (distal enhancer) and PE (proximal enhancer), are indicated. The Foxh1-binding sites in DE or PE are mutated in *L1-DE$^{m}$PE* and *L1-PE$^{m}$*, respectively. **c** The expression of *L1-DE+PE+-lacZ* was examined by X-gal staining in *Nodal+/+* and *Nodal−/−* embryos at the indicated stages. Scale bars, 50 μm. **d** *L1-DE$^{m}$PE+-Cre* or *L1-2.2-Cre* transgenic mice were crossed with *Rosa26R* transgenic mice, and transgenic embryos recovered at E5.5 or E6.5 were stained with X-gal. Two types of embryos were observed for the *L1-DE$^{m}$PE+-Cre* cross: type I (8/24 embryos), in which only DVE and DVE-derived cells were marked at E5.5 and E6.5, respectively; and type II (16/24 embryos), in which the extraembryonic region was positive in addition to DVE and DVE-derived cells at E5.5 and E6.5. DVE-derived cells were detected on the lateral side of E6.5 embryos produced from the *L1-2.2-Cre* cross (6/7 embryos). The number of DVE-derived cells was increased in E6.5 embryos produced from a cross of *L1-2.2-Cre* mice with *Lefty1,2−/−* mice expressing *Rosa26R* (2/3 embryos)

whereas the Foxh1-binding sites in PE regulate *Lefty1* expression in L1^dve cells at E4.5. DE may also contribute to the regulation of *Lefty1* expression at E4.5, given that the expression level of *L1-DE^mPE^+-lacZ* at this time (Supplementary Fig. 2d) was lower than that of *L1-DE^+PE^+-lacZ* (Fig. 1c) (note that the LacZ

staining time for the former embryo was 12 h, whereas that for the latter embryo was 15 min).

**Nodal signaling induces *Lefty1* expression.** Given that our results suggested that Nodal-Foxh1 signaling regulates *Lefty1*

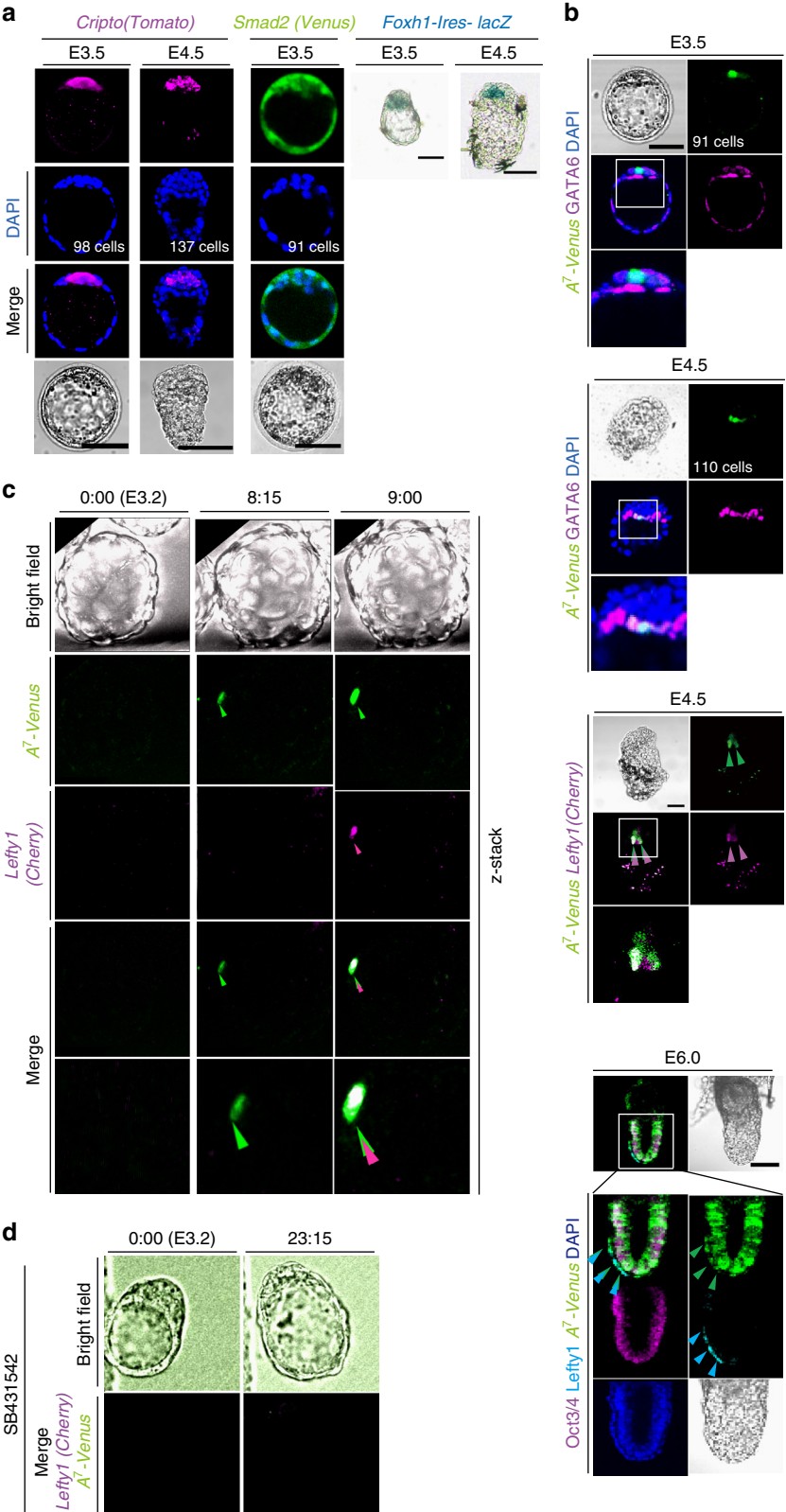

expression in L1$^{epi}$ and L1$^{dve}$ cells, we next examined expression of Nodal signaling components. Cripto (E3.5: 5/5 embryos; E4.5: 6/6 embryos), Foxh1(E3.5: 4/5 embryos; E4.5: 8/8 embryos), and Smad2(E3.5: 6/7 embryos) were all found to be expressed between E3.5 and E4.5 (Fig. 2a)[9]. Nodal-Foxh1 signaling activity can be monitored with a transgene driven by seven tandem repeats of a Foxh1-binding sequence[18]. When examined with such a transgene, A$_7$-Venus, Nodal-Foxh1 signaling was found to be active in a few ICM cells at E3.5 (presumably L1$^{epi}$ cells, 11/11 embryos), in a few GATA6$^+$ cells at E4.5 (L1$^{dve}$ cells, given that they also expressed Lefty1, 12/14 embryos), and in AVE and epiblast cells at E6.0, as expected (10/10 embryos) (Fig. 2b). Culture of E3.2 embryos for 24 h with SB431542 prevented A$_7$-Venus expression (11/11 embryos) (Supplementary Fig. 3a), whereas it was maintained in the control embryos (5/5 embryos), confirming that such expression is dependent on Nodal signaling. Simultaneous monitoring of both A$_7$-Venus and Lefty1 expression in live embryos from E3.2 revealed that a single cell became positive for Venus expression after culture for ~ 8 h (Fig. 2c, Supplementary Movie 1). This cell then began to express Lefty1. Lefty1$^+$ cells in such live imaging at this time (equivalent to E3.5) were positive for Venus (15/16 embryos), which is consistent with the notion that Lefty1 expression in L1$^{epi}$ cells is induced by Nodal signaling. Furthermore, expression of A$_7$-Venus and Lefty1 (Cherry) was abolished in the presence of SB431542 (Fig. 2d, Supplementary Movie 2). Lefty1$^+$ cells at E4.5 were also positive for Venus (7/7 embryos) (Fig. 2b), suggesting that Lefty1 expression in L1$^{dve}$ cells is also regulated by Nodal signaling. While Lefty1 expression was found in PrE at E4.5, Cripto was not expressed in PrE (Fig. 2a). It is most likely that PrE cells can receive Nodal siganing because Cryptic, another co-receptor for Nodal, is expressed in PrE[19], and because Cripto and Cryptic can act non-cell-autonomously[20,21]. In support of this, Lefty1$^+$ cells at E5.5 (DVE cells) are lost in the Cripto$^{-/-}$, Cryptic$^{-/-}$ double mutant embryo[19].

We next monitored Nodal and Lefty1 expression simultaneously in live embryos between E3.2 and E4.0 (Fig. 3a, Supplementary Movie 3). Nodal expression, as revealed with a Nodal(Tomato) BAC transgene, was dynamic, beginning in one cell, rapidly expanding to all cells, and being maintained until E4.0. Lefty1 expression (in L1$^{epi}$ cells) was found to begin in the same cell that had earlier initiated Nodal expression (12/24 embryos) or in a neighboring cell (11/24 embryos).

We also examined the effect of ectopic Nodal expression on Lefty1 expression. Injection of Nodal mRNA and mTomato mRNA into E3.2 embryos harboring the Lefty1(mVenus) BAC transgene and examination of the embryos 6 h later revealed that Lefty1 expression began either in the cell that received Nodal mRNA (6/19 embryos) or in a neighboring cell (12/19 embryos) (Fig. 3b–e). In the remaining embryo (1/19 embryos), Lefty1 expression was found in a distant cell. Injection of Nodal mRNA did not increase the number of Lefty1$^+$ cells at E4.5 (Fig. 3f and

Supplementary Fig. 3b, c). When Nodal mRNA and mTomato mRNA (encoding a membrane-localized form of Tomato) were injected into E3.2 embryos harboring the A$_7$-Venus transgene, Venus was detected in the cell that received Nodal mRNA or in a neighboring cell (Fig. 3g). We confirmed that Nodal protein was produced in the excess injected cell (Fig. 3h). When mTomato mRNA alone was injected, however, Venus$^+$ cells were located randomly relative to the mTomato$^+$ cell (Fig. 3d, e). Collectively, these results suggested that Lefty1 expression in L1$^{epi}$ and L1$^{dve}$ cells is induced by Nodal-Foxh1 signaling.

**Lefty activity restricts the number of prospective DVE cells.** We next investigated whether Lefty proteins might play a role in DVE formation. In addition to Lefty1, Lefty2 was also expressed in mouse embryos between E3.5 and E4.5. Expression of Lefty2, as revealed with a Lefty2(lacZ) BAC transgene, was thus detected in a subset of ICM cells at E3.5 (Supplementary Fig. 4a, b: 16/16 embryos). At E4.5, Lefty2 was expressed in a subset of PrE and EPI cells (Supplementary Fig. 4b). Lefty2 expression at E4.5 was also dependent on Nodal-Foxh1 signaling, given that such expression was not apparent in Nodal$^{-/-}$ (3/3 embryos) or Foxh1$^{-/-}$ (4/4 embryos) embryos (Supplementary Fig. 4a). Monitoring of Lefty2 and Lefty1 expression from E3.2 also revealed that both genes were expressed in the same ICM cells (L1$^{epi}$ cells) (6/10 embryos) (Supplementary Fig. 4b and Supplementary Movie 4) or in different ICM cells (4/10 embryos). Both genes also initiated their expression with similar timing (Supplementary Fig. 4c and Supplementary Movie 4). Given that Lefty2 was also found to be expressed in ICM cells at E3.5, we generated a mutant (Lefty1,2$^{-/-}$) mouse lacking both Lefty1 and Lefty2 (Supplementary Fig. 5a). Staining of Lefty1,2$^{-/-}$ embryos with antibodies that recognize both Lefty1 and Lefty2 confirmed the absence of Lefty proteins (5/5 embryos) (Supplementary Fig. 5b).

We then examined the effect of Lefty1 and Lefty2 deletion on the number of future DVE cells. Prospective DVE cells were identified and counted in Lefty1(mVenus) BAC transgenic embryos at E4.5 (Fig. 4a). The number of future DVE cells (mVenus$^+$ cells) was increased in the absence of Lefty genes (Fig. 4a, d, e, and Supplementary Table 2), whereas the number of PrE cells (Fig. 4b, Supplementary Table 2) or ICM cells (Fig. 4c, Supplementary Fig. 5, and Supplementary Table 2) was not significantly affected. mVenus$^+$ cells thus constituted ~ 10% of total PrE cells (GATA6$^+$ cells) in wild-type (WT) embryos, whereas they accounted for 25 to 30% of PrE cells in some Lefty1$^{-/-}$ embryos (Fig. 4e, Supplementary Table 2). This finding is consistent with our previous observation that the number of cells expressing Cerl1 (a marker for DVE) at E5.5 was increased in Lefty1$^{-/-}$ embryos[22]. Moreover, future DVE cells accounted for 30–50% of PrE cells in Lefty1,2$^{-/-}$ embryos (Fig. 4e, Supplementary Table 2). These results suggested that Lefty1 is not only a

---

**Fig. 2** Lefty1 expression begins in cells positive for Nodal-Foxh1 signaling. **a** Expression of Nodal signaling components in mouse embryos at E3.5 and E4.5. Expression of Cripto(Tomato) and Smad2(Venus) BAC transgenes as well as that of GATA6 were monitored by immunofluorescence staining, whereas that of a Foxh1-Ires-lacZ transgene was monitored by X-gal staining. Scale bars, 50 μm. **b** Embryos harboring A$_7$-Venus were examined for Venus and GATA6 immunofluorescence at E3.5 and E4.5 (top two panels). Note that Venus is expressed in a GATA6$^-$ cell at E3.5 and in GATA6$^+$ cells at E4.5. An E4.5 embryo harboring A$_7$-Venus and a Lefty1(Cherry) BAC transgene was examined for Venus and Cherry immune fluorescence (third panel). Arrowheads indicate two Lefty1$^+$ cells that are also positive for Venus. An E6.0 embryo harboring A$_7$-Venus was subjected to immunofluorescence staining for Venus, Lefty, and Oct3/4 (bottom panel). Arrowheads indicate Cherry in magenta, Venus in green and Lefty1 in light blue. Boxed regions are shown at higher magnification in the images immediately below. Scale bars, 50 μm. **c** An E3.2 embryo harboring both Lefty1(Cherry) and A$_7$-Venus transgenes was cultured for 9 h. Fluorescence of Venus and Cherry was examined at the indicated times. A single cell positive for Venus was detected at 8.25 h (arrowhead). This cell had being positive for Cherry at 9 h. **d** An E3.2 embryo harboring both Lefty1(Cherry) and A$_7$-Venus transgenes was cultured for 23.15 h in the presence of 10 μM SB431542 and monitored for Venus and Cherry fluorescence. Note that Venus or Cherry fluorescence is not apparent at 23.15 h

marker for future DVE cells in the blastocyst, but that it also restricts the number of prospective DVE cells together with Lefty2. Although the number of prospective DVE cells was increased in Lefty1,2$^{-/-}$ embryos at E4.5, the expression patterns of Cerl1 and Hex appeared normal at E6.5 (9/9 embryos for Cerl1, 6/6 embryos for Hex) and E7.5 (5/5 embryos for Cerl1) (Supplementary Fig. 5d, e), suggesting that AVE is formed normally even though such AVE cells lack Lefty1. Since Cerl1 is required to position the primitive streak at the posterior side of the embryo[23], together with Lefty1 and Lefty2, Cerl1 may have a redundant role in AVE formation.

**Nodal-Lefty regulatory network.** Our results indicated that Lefty1 expression is induced by Nodal signaling and that Lefty1

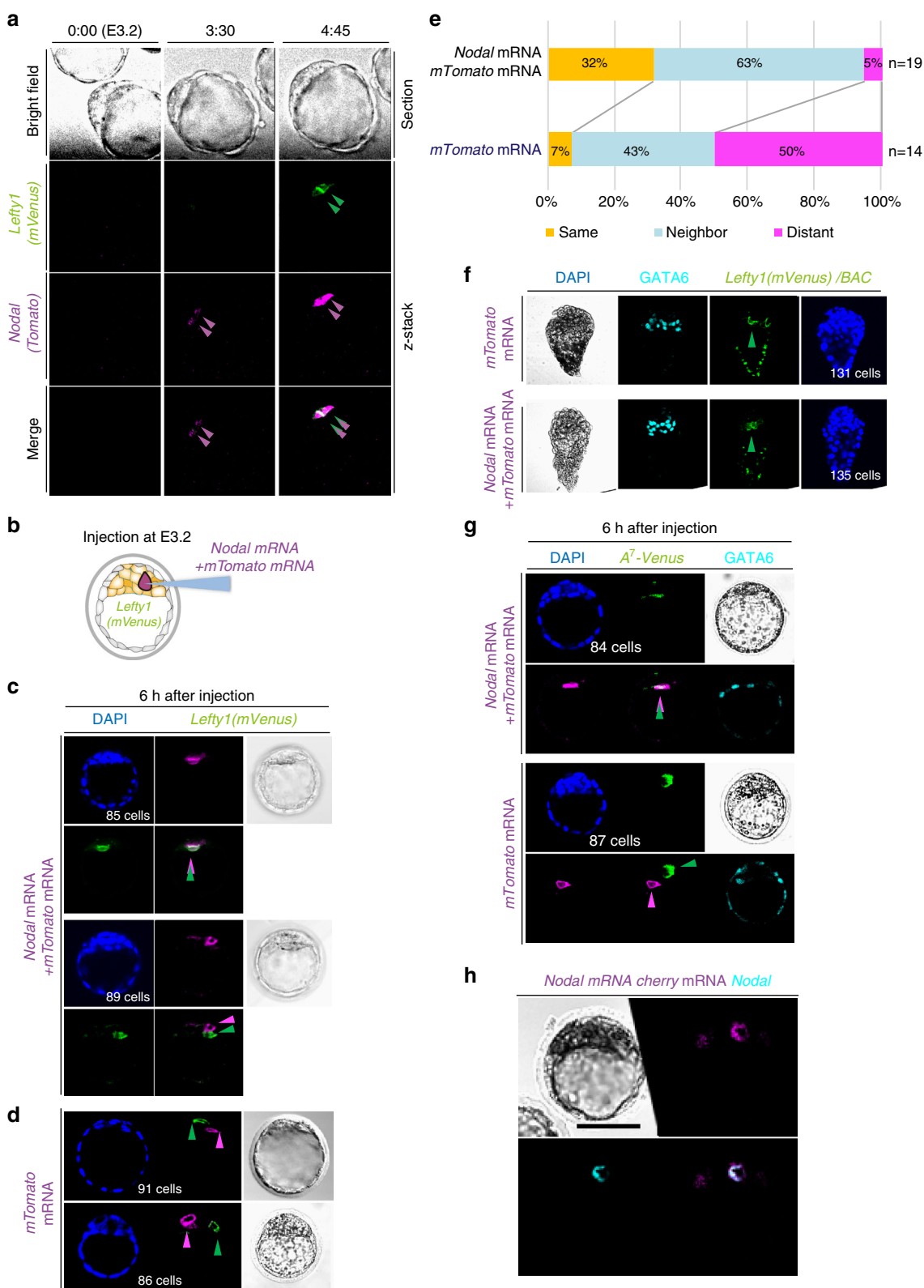

restricts the number of Lefty1$^+$ cells, a scenario reminiscent of the self-enhancement and lateral inhibition (SELI) Nodal-Lefty regulatory network that operates during L–R patterning at E8.0[24]. Given that Lefty proteins diffuse faster than Nodal[25], whether a Nodal-Lefty SELI system operates in peri-implantation embryos will depend on whether Nodal expression is positively regulated by Nodal itself. We therefore investigated how Nodal expression is regulated at the blastocyst stage—in particular, whether it is positively autoregulated as it is at E8.0. Transcription of Nodal is regulated by several enhancers, including two Foxh1-dependent enhancers, ASE[26,27] and LSE[27], which contribute to L–R asymmetric expression of Nodal at E8.0 (Fig. 5a) and which are also active at the peri-implantation stage[28]. Another enhancer, PEE, whose activity depends on Wnt–β-catenin signaling, is also active in a subset of cells in the blastocyst[28].

We found that ASE, LSE, and PEE all contribute to the regulation of Nodal expression at E4.5. Deletion of ASE thus attenuated Nodal expression (14/15 embryos) (Fig. 5b). Whereas additional deletion of LSE did not have a further substantial effect on Nodal expression (17/19 embryos), which of both LSE and PEE did further reduce it (7/10 embryos) (Fig. 5b). Given that ASE and LSE are dependent on Foxh1 and that PEE is dependent on Wnt–β-catenin, these results suggested that Nodal expression in peri-implantation mouse embryos is positively regulated by Nodal signaling. Consistent with this notion, culture of E3.2 embryos with SB431542 for 24 h resulted in marked attenuation of Nodal expression, presumably in L1$^{epi}$ and L1$^{dve}$ cells (13/13 embryos) (Fig. 5c). Furthermore, Nodal expression in L1$^{dve}$ cells at E4.5 was greatly reduced in Foxh1$^{-/-}$ embryos (4/4 embryos) (Fig. 5d). Together, these observations suggested that Nodal expression in peri-implantation embryos is positively regulated by Nodal signaling via Foxh1-dependent enhancers (Fig. 5e).

**L1$^{epi}$ and L1$^{dve}$ cells are selected randomly**. Injection of Nodal mRNA into a single cell at E3.2 induced Lefty1 expression in the same cell, generating an L1$^{epi}$ cell (Fig. 3c), but it did not increase the overall number of L1$^{epi}$ cells (Supplementary Fig. 3b, c). When such injected embryos were returned to the uterus and allowed to develop further, they developed normally at least up to E6.5 (5/5 embryos), showing normal expression patterns of Cerl1, Lefty1, and Lefty2 (Fig. 6a). Both Cerl1 and Lefty1 were thus detected at the anterior side, suggesting that the AVE was formed normally. Lefty2 was expressed at the opposite (posterior) side, suggesting that the primitive streak was formed correctly. Examination of the fate of the injected cell revealed that it survived and contributed to DVE and DVE-derived cells at E6.5 (Fig. 6b). We also removed all (usually two or three) Lefty1$^+$ cells

from Lefty1(mVenus) transgenic embryos at E3.5 by laser ablation. Cell ablation was confirmed by monitoring of the cell membrane, with successful ablation resulting in rupture of the membrane and the appearance of fluorescent cell debris (Fig. 6c). Culture of the ablated embryos revealed that Lefty1 was expressed in a different cell by 8 h after the ablation (8/8 embryos) (Fig. 6c). On return to the uterus, such ablated embryos again developed an apparently normal A–P axis by E6.5, with Lefty1 and Cerl1 expression being apparent on the anterior side of the embryo and Lefty2 expression on the posterior side (10/10 embryos) (Fig. 6d).

Similarly, when all (usually four or five) L1$^{dve}$ cells at E4.0 were ablated, a new Lefty1$^+$ cell appeared in the PrE region by 8 h after the ablation (Fig. 7a). On return of such embryos to the uterus, they developed an apparently normal A–P axis by E6.5 (Fig. 7b).

Together, these results suggested that both L1$^{epi}$ and L1$^{dve}$ cells are not predetermined but are selected in a regulated and robust manner. Furthermore, the A–P axis can be established normally even if an ectopic cell begins to express Lefty1 between E3.5 and E4.0; that is, an ectopic cell can be specified to become a prospective DVE cell.

## Discussion

Lefty1 is expressed first in L1$^{epi}$ cells at E3.5 and subsequently in L1$^{dve}$ cells. Although Lefty1 expression specifically marks future DVE cells in the mouse embryo at E4.0 to E4.5, it is not absolutely required to specify DVE cells, given that Cerl1$^+$Hex$^+$ cells are formed at E6.5 in the absence of both Lefty1 and Lefty2. Rather, Lefty1 and Lefty2 function to restrict the number of future DVE cells. Thus, in the absence of either Lefty1 alone or both Lefty1 and Lefty2, the number of future DVE cells is increased at E4.5. The number of DVE cells at E5.5 (Cerl1-expressing cells) was previously shown to be increased in Lefty1$^{-/-}$ embryos[22]. However, we found that the number and position of AVE cells at E6.5 (Cerl1-expressing cells) remained essentially normal in Lefty1,2$^{-/-}$ embryos. Although DVE guides the anterior migration of AVE cells that arise later, the increased number of DVE cells in the mutant embryos does not appear to influence the appearance and migration of AVE cells.

Then, why does Lefty1 expression occur in two steps? In particular, what is the role of the first phase of Lefty1 expression in L1$^{epi}$ cells? Lefty1 expression in L1$^{epi}$ cells is transient, disappearing shortly after L1$^{dve}$ cells arise[8]. Lefty1 produced in L1$^{epi}$ cells may establish an uneven distribution of Nodal activity within the blastocyst, as revealed by the pattern of A$_7$-Venus expression at E3.5 (Fig. 2c), and thereby allow only one or two cells to become L1$^{epi}$ cells (Fig. 7c). This would in turn restrict the number of L1$^{dve}$ cells and determine the position of these cells in the blastocyst. In support of this notion, L1$^{dve}$ cells arise in a

**Fig. 3** Lefty1 expression in L1$^{epi}$ cells is induced by Nodal. **a** An E3.2 embryo harboring both Lefty1(mVenus) and Nodal(Tomato) BAC transgenes was cultured for 5 h, with fluorescence of mVenus and Tomato being examined at the indicated times. Nodal expression began in two cells at 3.5 h (magenta arrowheads), with these two cells becoming positive for Lefty1 expression by 4.75 h (green arrowheads). **b** Schematic illustration of mRNA injection experiments in (**c**) and (**d**). mTomato mRNA (encoding a membrane-localized Tomato) was injected with or without Nodal mRNA into a single cell of E3.2 embryos harboring the Lefty1(mVenus) BAC transgene. Embryos were examined for mTomato and mVenus immunofluorescence 6 h after mRNA injection. **c** Embryos injected with both mTomato and Nodal mRNAs. mVenus is expressed in the injected cell (upper panel) or in a neighboring cell of the injected cell (lower panel). **d** Two embryos injected with mTomato mRNA alone. mTomato$^+$ and mVenus$^+$ cells (arrowheads) do not overlap. **e** Summary of the location of the mVenus$^+$ cell relative to the injected cell for experiments similar to that in **c** and **d**. **f** E3.2 embryos harboring the Lefty1(mVenus) BAC transgene were injected with mTomato mRNA alone or together with Nodal mRNA, allowed to develop in utero, recovered at E4.5, and examined for Lefty1(mVenus) and GATA6 expression. Note that there is no substantial difference in the number of Lefty1-expressing cells between the two types of injected embryos (Supplementary Fig. 3b, c). **g** E3.2 embryos harboring A$_7$-Venus were injected with mTomato mRNA alone or together with Nodal mRNA and were examined for GATA6 and Venus expression after 6 h. When mTomato mRNA alone was injected, the Venus$^+$ cell was randomly located relative to the injected cell (lower panel). However, when Nodal mRNA was co-injected, the injected cell or a neighboring cell was positive for Venus (upper panel). **h** An E3.2 embryo co-injected with excess Nodal mRNA (200 ng/μl) and with mTomato mRNA was immunostained for Nodal at 6 h after mRNA injection. Scale bar, 50 μm

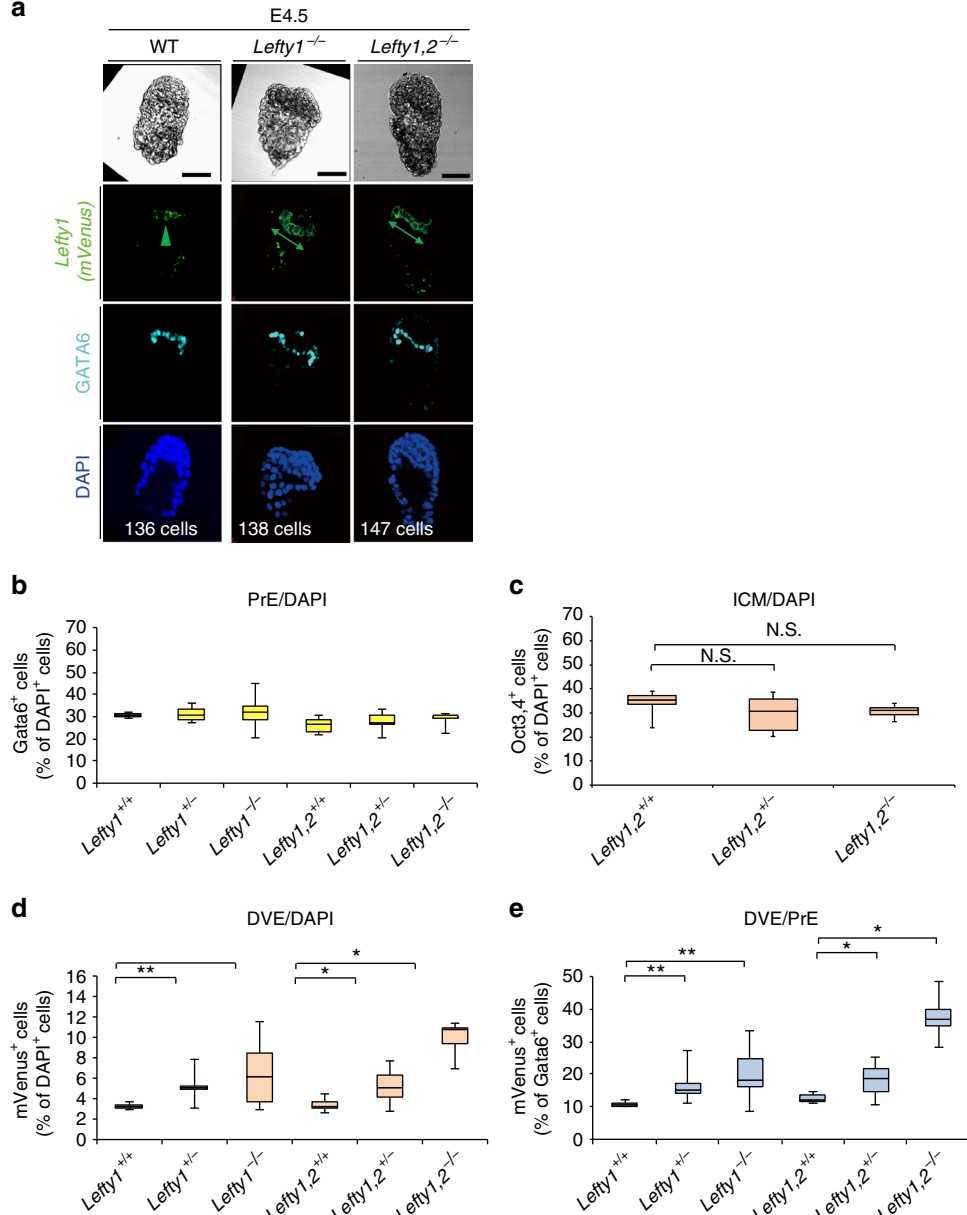

**Fig. 4** Lefty1 and Lefty2 restrict the number of prospective DVE cells. **a** The number of prospective DVE cells [GATA6$^+$ cells expressing the *Lefty1(mVenus)* BAC transgene] was examined in WT, *Lefty1$^{-/-}$*, and *Lefty1,2$^{-/-}$* embryos at E4.5. The embryos were subjected to immunofluorescence staining for mVenus and GATA6. Scale bars, 50 μm. Arrowheads and double-headed arrows indicate mVenus$^+$ cells (prospective DVE cells). **b**-**e** Summary of the number of PrE cells (GATA6$^+$ cells) as a percentage of total cells (**b**), the number of ICM cells positive for Oct3/4 staining as a percentage of total cells (**c**) (Supplementary Fig. 5c), the number of prospective DVE cells as a percentage of total cells (**d**), and the number of prospective DVE cells as a percentage of total PrE cells (**e**) in embryos of the indicated genotypes at E4.5. Data are presented as box-and-whisker plots (first and third quartile, the line represents the median; whiskers: minimum to maximum). *$P < 0.05$, **$P < 0.01$; NS, not significant (*t* test)

region remote from L1$^{epi}$ cells[8]. Unfortunately, it is not technically feasible to address the role of Lefty1 in L1$^{epi}$ cells directly, given that a Cre transgene active specifically in E3.5 epiblasts is not currently available (the expression of zygotic *Sox2-Cre*[29] is not sufficiently early). Nonetheless, it is of note that the number of prospective DVE cells was increased to a markedly greater extent in *Lefty1,2$^{-/-}$* embryos than in *Lefty1$^{-/-}$* embryos. Whereas *Lefty2* and *Lefty1* are both expressed in epiblast-fated cells at E3.5 (Supplementary Fig. 4a–c), they are expressed in different domains at E4.5, suggesting that Lefty1 and Lefty2 in epiblast-fated cells at E3.5 regulate the number of future DVE cells.

Ablation of L1$^{epi}$ or L1$^{dve}$ cells at E3.5 and E4.0, respectively, resulted in the initiation of *Lefty1* expression in remaining cells that presumably took over the role of the ablated cells. Injection of *Nodal* mRNA into a single cell of E3.2 embryos induced *Lefty1* expression, but did not increase the number of L1$^{epi}$ cells. Together, these results suggest that selection of both L1$^{epi}$ and L1$^{dve}$ cells is random and regulated (Fig. 7c). Consistent with this notion, the positions of L1$^{epi}$ cells at E3.5 and of L1$^{dve}$ cells at E4.0 appear random, although L1$^{dve}$ cells eventually occupy the future anterior side of the PrE at E4.5. Furthermore, embryos with *Nodal* mRNA injected into a single cell developed a normal A–P axis at E6.5, suggesting that the A–P axis is established

normally even if an ectopic cell is chosen to become an L1$^{epi}$ or L1$^{dve}$ cell. Overall, our data suggest that selection of prospective DVE cells is both random and regulated, and that there may be no fixed prepattern for the future A–P axis before the blastocyst stage, although some cellular asymmetries along the future A–P axis have been detected around E5.5 (refs. [30,31]). Further

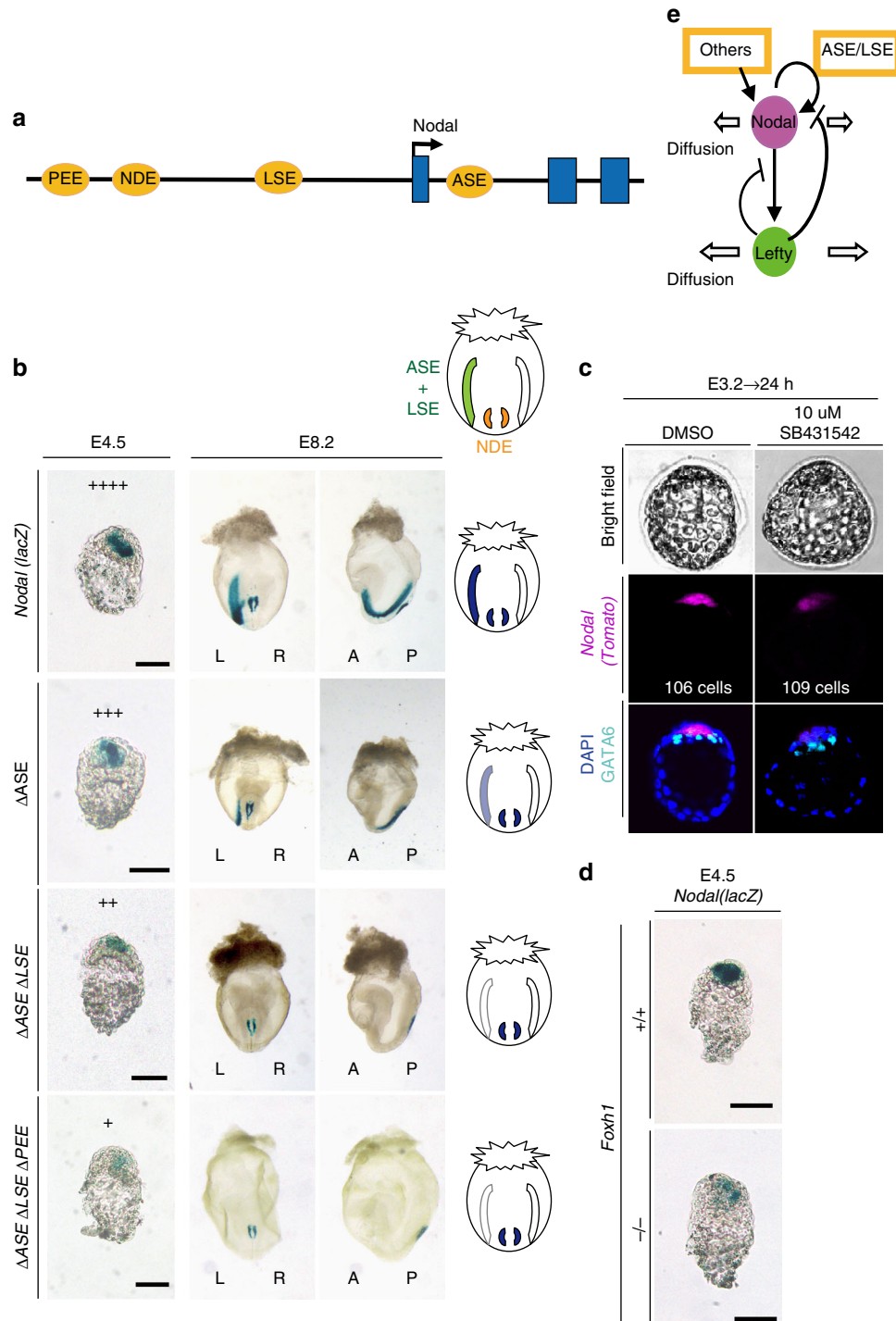

**Fig. 5** Regulation of *Nodal* expression in peri-implantation embryos. **a** Four transcriptional enhancers—ASE, LSE, NDE, and PEE—contribute to regulation of *Nodal* expression. **b** Expression of *Nodal(lacZ)* BAC transgenes containing all enhancers or lacking either ASE alone, ASE and LSE, or ASE, LSE, and PEE was determined in embryos at E4.5 and E8.2 by X-gal staining. Scale bars, 50 μm. In E4.5 embryos, the relative levels of *Nodal* expression in the different embryos are indicated by plus signs. Frontal and lateral views are shown for E8.2 embryos. X-gal$^+$ regions are schematically summarized in diagrams on the right. Note that ASE and LSE are active in the lateral plate mesoderm, whereas NDE is active in the node[40]. **c** E3.2 embryos harboring a *Nodal(Tomato)* BAC transgene were cultured for 24 h with 10 μM SB431542 or dimethyl sulfoxide (DMSO) vehicle and were then examined for Tomato and GATA6 immunofluorescence. **d** Expression of the *Nodal(lacZ)* BAC transgene in *Foxh1$^{+/+}$* and *Foxh1$^{-/-}$* embryos at E4.5. Scale bars, 50 μm. **e** Regulatory relation between *Nodal* and *Lefty* at the peri-implantation stage. The genes constitute a SELI system, as they do for L–R patterning at E8.0

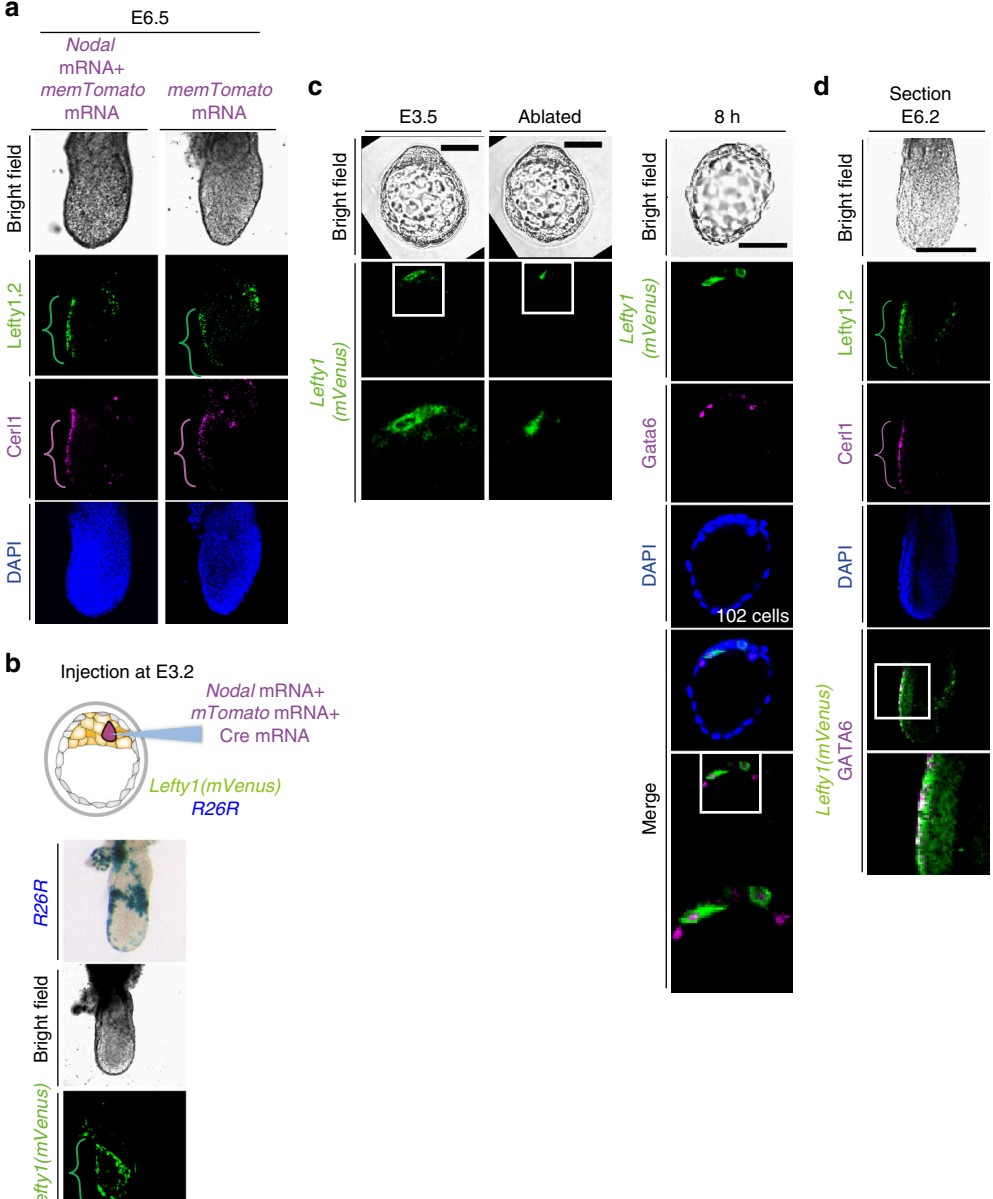

**Fig. 6** Ablation of L1^epi cells at E3.5 results in the appearance of new L1^epi cells and does not impair A–P patterning. **a** E3.2 embryos in which a single cell had been injected with *mTomato* mRNA alone or together with *Nodal* mRNA were transferred to a pseudopregnant mouse, allowed to develop until E6.5, and then recovered for immunofluorescence staining of A–P markers. The brackets denote AVE located on the anterior side of the embryos. Lefty1, Lefty2, and Cerl1 were detected in AVE, the primitive streak, and AVE, respectively, suggesting that a normal A–P axis was established. **b** An E3.2 *Rosa26R* (*R26R*) embryo harboring a *Lefty1(mVenus)* BAC transgene was injected with *Nodal* mRNA, *mTomato* mRNA, and *Cre* mRNA, transferred to a pseudopregnant mouse, allowed to develop until E6.5, and then recovered for X-gal staining and immunofluorescence staining for mVenus. Note that the injected cell survived and contributed to DVE and DVE-derived cells. **c** All mVenus^+ cells (L1^epi cells) were removed by laser ablation from an E3.5 embryo harboring a *Lefty1(mVenus)* BAC transgene, after which the embryo was cultured for 8 h and then subjected to immunofluorescence staining for mVenus and GATA6. Note that fluorescent membrane debris was detected immediately after ablation and that new mVenus^+ cells had appeared by 8 h after the ablation. Boxed areas are shown at higher magnification in the images immediately below. Scale bar, 50 μm. **d** After culture, an embryo such as that in (**c**) was transferred to a pseudopregnant mouse, allowed to develop until E6.5, and then recovered for immunofluorescence staining of A–P markers. Lefty1, Lefty2, and Cerl1 were detected in AVE, the primitive streak, and AVE, respectively. Scale bar, 100 μm

understanding of the origin of the A–P axis in mammals may require investigation of the origin and regulation of Nodal activity during earlier stages.

## Methods

**Mice.** Various *Lefty1* BAC transgenes were constructed from the mouse *Lefty1* BAC clone RP23-390I1. The *Lefty1(lacZ)* BAC and *Lefty1(mVenus)* BAC transgenes were described previously[8,9]. *Lefty1(Cherry)* BAC and *Lefty2(mTomato)*

BAC transgenes were similarly constructed. A transgenic mouse line harboring *L1-DE^+PE^+-lacZ* was previously established[9]; this transgene recapitulates asymmetric *Lefty1* expression in PrE and AVE. *L1-mVenus*, *L1-Venus* and *L1-Cre* transgenes were constructed by replacing *lacZ* of *L1-DE^+PE^+-lacZ* and related constructs with *mVenus* or *Venus*. The *A₇-Venus* transgene contains seven tandem repeats of a Foxh1-binding sequence as well as an *Hsp68-Venus* hybrid gene. A *Lefty2(lacZ)* BAC transgene was constructed from the mouse *Lefty2* BAC clone RP23-390I1 (the same clone as for *Lefty1* above) by replacement of exon 1 with *lacZ*. A *Smad2(Venus)* BAC transgene was constructed by replacement of exon 1 of *Smad2* in the mouse BAC clone RP23-90N19 with *Venus*. *Cripto(Tomato)* and

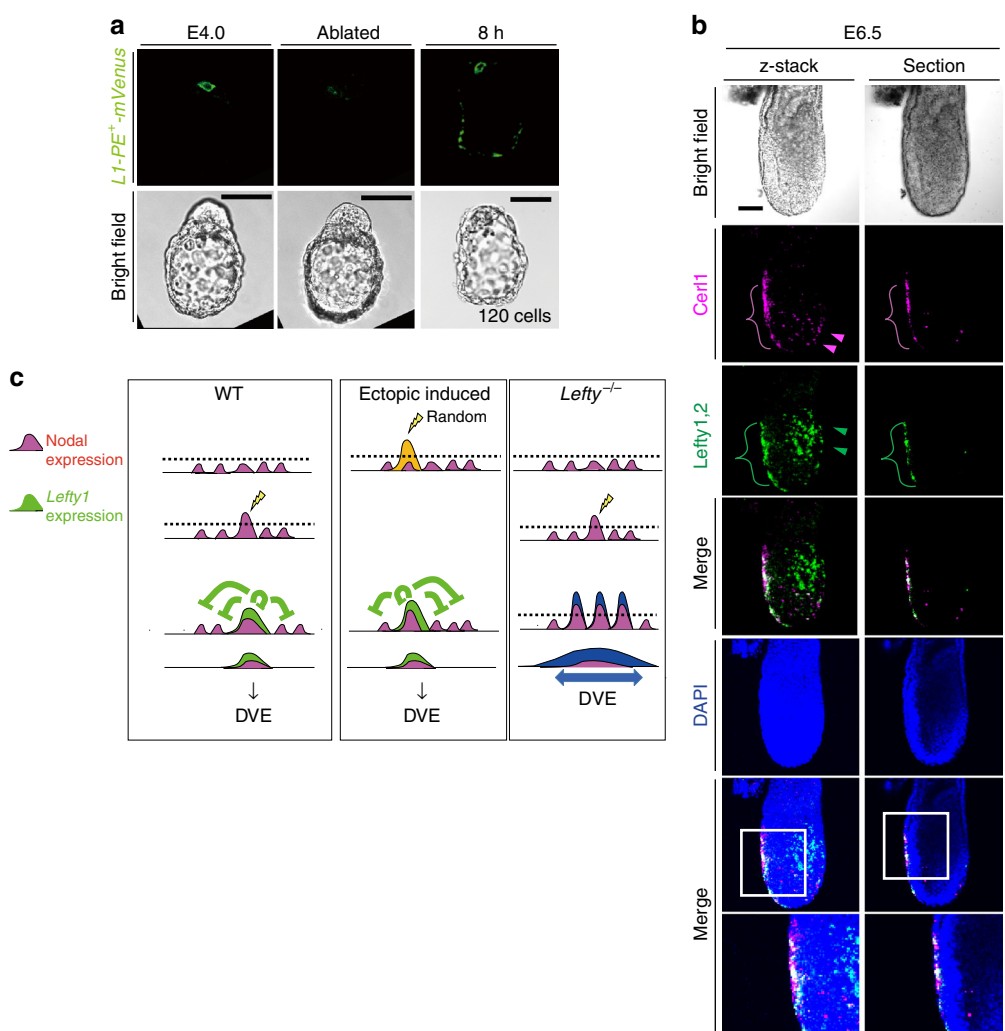

**Fig. 7** Ablation of L1$^{dve}$ cells at E4.0 results in the appearance of new L1$^{dve}$ cells and does not impair A–P patterning. **a** All mVenus$^+$ cells (L1$^{dve}$ cells) were removed by laser ablation from an E4.0 embryo harboring an L1-PE$^+$-mVenus transgene (Fig. 1b), after which the embryo was cultured for 8 h and then subjected to immunofluorescence staining for mVenus. Note that a new mVenus$^+$ cell had appeared by 8 h after the ablation. Scale bars, 50 μm. **b** After culture, an embryo such as that in **a** was transferred to a pseudopregnant mouse, allowed to develop until E6.5, and then recovered for immunofluorescence staining of A–P markers. Lefty1 and Cerl1 were detected in the AVE, suggesting that a normal A–P axis was established. Arrowheads indicate Cerl1$^+$ definitive endoderm cells in magenta, and Lefty2$^+$ definitive ectoderm cells in green. The boxed regions of the merged images are shown at higher magnification in the images immediately below. **c** Model for the spatial distribution of Nodal expression and Lefty1 expression in a peri-implantation embryo based on observations in the present study. In the WT embryo (left panel), a cell that first expresses Nodal beyond a threshold level (dotted line) begins to express Lefty1. The Lefty1 protein then produced rapidly represses Nodal signaling in nearby cells, preventing them from expressing Lefty1. Injection of Nodal mRNA into a cell (middle panel) induces expression of Lefty1. In the absence of Lefty (right panel), a larger number of cells manifest Nodal expression or activity beyond the threshold and are fated to become DVE. See text for further details

Nodal(Tomato) BAC transgenes were constructed from RP23-322L8 and RP23-55A6, respectively, by replacement of exon 1 of each gene with tdTomato. Nodal (lacZ) BAC transgenes lacking either ASE alone, ASE and LSE, or ASE, LSE, and PEE were constructed from a Nodal(lacZ) BAC[32]. An Oct3/4(mTomato) BAC transgene was constructed by insertion of the coding sequence for mTomato after the initiation codon of the mouse Oct3/4 BAC clone RP23-38P5. Recombinant BAC clones were generated with the use of the highly efficient recombination system for Escherichia coli[33]. BAC DNA was prepared by CsCl centrifugation and was linearized before microinjection[34]. Transgenic mice were generated as described previously[15]. Other mice used in this study have been reported: Lefty1$^{+/-}$ (ref. [35]), Foxh1$^{+/-}$ (ref. [36]), Nodal$^{+/-}$ (ref. [37]), Nodal(lacZ) BAC[32], and Foxh1-Ires-lacZ BAC[38]. Lefty1,2$^{+/-}$ mice were generated as described in Supplementary Fig. 5a. All transgenic mice were generated in C57BL6 and C3H F1 hybrid mice, whereas Lefty1,2$^{+/-}$ mice were under the C57BL/129 mixed background. All mouse experiments were approved by the relevant committees of Osaka University and RIKEN Center for Developmental Biology, license numbers FBS-12-019 and AH28-01.

**X-gal staining**. Transgenic embryos were stained with the X-gal (5-bromo-4-chloro-3-indolyl-β-D-galactopyranoside) substrate[26].

**Immunofluorescence analysis**. Embryos were recovered in phosphate-buffered saline (PBS) and staged on the basis of their morphology. They were fixed for 15 min at room temperature in PBS containing 4% paraformaldehyde, washed twice with PBS, permeabilized for 20 min at room temperature with 0.2% Triton X-100 in PBS, incubated first for 1 h at room temperature with TSA blocking reagent (Perkin-Elmer) and then overnight at 4 °C with primary antibodies diluted in blocking reagent. They were then washed three times with PBS before incubation with secondary antibodies diluted in blocking reagent. Nuclei were stained by incubation for 30 min at room temperature with DAPI (1/2000 dilution in PBS) (Wako). All images were acquired with the use of a laser-scanning confocal microscope system (FV1000, Olympus) and a UPLSAPO 20× objective lens (numerical aperture, 0.75; Olympus). Primary and secondary antibodies applied for immunofluorescence staining are listed in Supplementary Table 1.

**Injection of mRNA**. Capped synthetic mRNAs encoding Nodal, mTomato and Cre were generated by in vitro transcription from the Nodal/pSP64T, mTomato/PCS2 and Cre/PCS2 vectors with the use of an SP6 mMessage mMachine Kit (Ambion AM1340). *Nodal* (35 ng/µl) and *mTomato* (100 ng/µl) mRNAs were introduced into a single cell of E3.2 embryos with the use of a Piezo-expert microinjector (Eppendorf). Injected embryos were cultured for 1.5 h and then checked with an M205FC fluorescence stereomicroscope (Leica), with only those with a single mTomato-positive cell being studied further.

**Whole-mount in situ hybridization**. Whole-mount in situ hybridization was performed according to standard procedures[39] with digoxigenin-labeled riboprobes specific for *Cerl1* or *Hex*. Embryos were genotyped by polymerase chain reaction analysis of partially purified embryonic DNA.

**Time-lapse microscopy and image processing**. Peri-implantation embryos were recovered in modified Whitten's medium[8] and then transferred to glass-bottom culture dishes (Mat Tk, P35G-0-14-C) in fresh medium for culture in a $CO_2$ incubator. Time-lapse images were obtained with a Cell Voyager CV1000 CSU confocal system (Yokogawa). The images were acquired from multiple positions at 15-min intervals and 3 µm apart in the z-axis for optical sectioning with a 20× objective lens (Olympus UplanApo; numerical aperture, 0.70). Confocal images were processed with IMARIS (Bitplane) for analysis of cell behavior.

**Laser ablation of cells**. All mVenus[+] cells in E3.5 embryos harboring *Lefty1* (*mVenus*) (Fig. 6) or in E4.0 embryos harboring *L1-PE[+]-mVenus* were ablated with the use of a TCS SP5 multiphoton microscope (Leica). The embryos were immersed in Whitten's medium containing 20 mM Hepes (pH 7.2). Membrane fluorescence of target cells was scanned in the confocal scanning mode with a 488-nm argon laser. Target cells were ablated with a pulsed TiSa 1-W 800-nm laser at full power in the two-photon mode for 1 to 2 s. The IR laser beam was restricted to the center of the cytoplasm of target cells in the region of interest (ROI) scan mode to prevent bleaching of membrane fluorescence and damage to neighboring cells. Successful ablation was confirmed by detection of changes to the cell membrane in the confocal scanning mode with the 488-nm argon laser. Ablated cells were thus identified by the presence of debris at the cell membrane (Fig. 6c). All images were acquired with a Leica HCX APO 20 × water-immersion objective lens (numerical aperture, 1.0).

**Statistical analysis**. The data were analyzed with t test. A P value of <0.05 was considered statistically significant and a P value of <0.01 was highly significant.

**Data availability**. The authors declare that all data supporting the findings of this study are available within the article and its Supplementary Information files or from the corresponding author upon reasonable request.

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

## Acknowledgements

We thank A. Fukumoto, Y. Uegaki, K. Ohnishi-Sugimura, E. Kajikawa and K. Miyama for technical assistance. This work was also supported by a grant from CREST (Core Research for Evolutional Science and Technology) of JST as well as by a Grant-in-Aid from the Ministry of Education, Culture, Sports, Science, and Technology of Japan (no. 17H01435). K.T. was supported by KAKENHI Grants-in Aid for Scientific Research on Innovative Areas (no. 15H01511 and 24116706) from the Japan Society for the Promotion of Science and KAKENHI Grants-in Aid for Young Scientists (A) (no. 24687028).

## Author contributions

The project was planned, the experiments were designed, and the manuscript was written by K.T. and H.H. Most experiments were performed and the data analyzed by K.T. Transgenes were constructed by H.N.

## Additional information

**Competing interests:** The authors declare no competing financial interests.

