## [Peer Review File · Nature Communications]

Reviewers' Comments:

Reviewer #1 (Remarks to the Author)

In this manuscript Takaoka and Hamada propose that future distal visceral endoderm cells (DVE) are already selected at blastocyst stages within the primitive endoderm (PrE).

Using live and static imaging, as well as some rather elaborate transgenic studies, the authors examine the role and regulation of Lefty1 in early DVE specification. They show that Lefty1 is expressed in a subset of epiblast cells (L1epi) at E3.5 and later in a subset of PrE cells (L1DVE) at E4.5.

The authors also show that this Lefty1 expression is dependent on Nodal-Foxh1. Lefty1 expression is regulated by two enhancers, designated as distal (DE) and proximal enhancer (PE) both containing FoxH1 binding sites. Analysis of lacZ reporter expression of a variety of Lefty1 transgenes containing both (DE and PE) enhancers, only the PE or mutated DE or PE, respectively, leads the authors to conclude that the PE drives Lefty1 expression in prospective DVE cells, and the DE in epiblast cells.

Further evidence for a Nodal dependent Lefty1 expression was obtained by injection of Nodal mRNA into (individual?) ICM cells induced Lefty1 expression, whereas inhibition of Nodal signaling prevented Lefty1 expression and led to an increase in DVE cells, as did the deletion of Lefty1, and Lefty1 and Lefty2. However neither of these scenarios affected anterior-posterior (A-P) axis formation which is a functional consequence of the migration of the DVE. A-P axis establishment was also apparently unaffected through the ablation of L1DVE cells blastocyst stage embryos.

Collectively these data lead the authors to posit that no fixed pre-pattern of the future A-P axis exists in the blastocyst, and that the selection of L1DVE cells within any given embryos is random and regulated by Nodal activity. These are important open questions that are difficult to address, but which lie at the heart of the regulative nature of the mammalian embryo. However, in this reviewer's opinion the conclusions drawn, and the proposed model are not well supported by the data which are generally not rigorously analyzed.

General comments:

- The resolution of the analysis presented in this study needs to be improved. High magnification views of ICMs should be shown in the figures.
- Blastocyst stages should be classified by cell numbers rather than by embryonic (E) day.
- Throughout the paper LacZ expression in blastocysts is hardly visible in the chromogenic stains, and should be rather analyzed with anti-LacZ antibodies which provide single cell resolution. Blastocysts should be counterstained with DAPI or co-stained with ICM and PrE specific markers for improved resolution of analysis.
- In many instances resolution of data presented does not allow one to distinguish between various scenarios and outcomes.
- Data should be analyzed more quantitatively.

Major Comments:

Figure 1:

a: lacZ staining in embryos is hardly visible.

The localization of the reporter (using antibodies against lacZ or Venus) should be shown using fluorescence, so that embryos can be co-stained with ICM (Nanog) and PrE (Gata6) specific markers.

b: Authors should put embryos expressing the various transgenic constructs into culture in the presence of inhibitors of Nodal signaling and vs. Nodal, and determine the effects on the population within the ICM.

Figure 2:

a: lacZ should be analyzed by IF. Smad2 expression is not distinguishable from background stain in this panel. Authors should use a pSmad2 antibody.

b: Authors should show high magnification views of these stages as well as intermediate stages and merges of the A7-Venus and Cer11 and Lefty1,2, respectively.

c: Single-plane of section, not rendering should be shown for the brightfield channel.

It would be interesting to see the expression of A7-Venus and Lefty1-Cherry in the presence of Nodal vs Nodal inhibitors.

Figure 3:

b: Stage of blastocyst needs to be defined better.

c: If overlayed on the brightfield channel the cherry positive cell is not ICM! Authors should show merge of DAPI channel.

c and d: Authors should include DAPI channel to show all cells and distinguish who is a neighbor, and who is not, and include histograms of cell numbers.

Figure 4:

a: Authors should include the cell number for each stage, otherwise it is very difficult to compare between stages.

b: Authors should include some statistics on the data, since if it weren't for one data point at 50 in Lefty1,2 -/-, Lefty1-/- and Lefty1,2-/- may be the same. In addition, can the authors comment on if the total number of Gata6 + cells were increased or the same. Perhaps data should be plotted as box plot?

Figure 5:

b: Numbers should be provided in this section for: embryos analyzed, embryos showing a defect, cells in embryos that are affected. Furthermore better quality images should be used, reviewer could not make out what is going on in the images of E4.5 embryos.

c: this reviewer is not sure if the tomato expression corresponds with the localization of the ICM in this panel.

In addition, what happens if embryos are cultured in Nodal?

d: Could authors include earlier stages and give numbers of embryos analyzed?

Figure 6:

The ablation experiments in theory are nice, but the authors do not show any controls for these studies.

How can the authors tell that they ablated and not bleached an mVenus cell?

Cell death markers to distinguish bleaching vs. ablating should be included.

Numbers of cells in ICM before and after ablation (to confirm cell ablation) should be included.

How do embryos look like immediately after ablation? Are any mVenus + cells left in the ICM after ablation?

Ablation experiments should be better explained in the text or methods. Authors should state in the text that these are laser ablations, not manual ablations!

Authors should clarify which laser or scope they used for the ablation. In the methods section they state the use of a two- or multi-photon scope and the 488nm laser, however, this laser line is not multi-photon, but a standard confocal line. A two-photon laser would be an IR laser which for ablating Venus or GFP would be in the 800 or 900nm range.

Authors should clarify if cell are ablated in only a single z-plane or in a total z-stack, hitting more than one cell.

Figure 7:

a and b should go into figure 6 and as panels presented in figure 6 need to be better control for. c: Authors have assessed Nodal expression, not the activity, if they want to discuss the activity they need to use the FoxH1 A7- transgene or pSmad2 localization.

Minor points:

Overall, the manuscript should be checked for typos and wording. Movies should be edited to draw the reader/viewer's attention to specific details.

Introduction:

Line 44-46: sentence does not make sense. "We know how the A-P axis is established at the level of the DVE. The questions should be how DVE is specified."

Line 46-48: sentence contradicts previous sentence.

Material and methods:

Time lapse microscopy and image processing:

Authors should include information on z-intervals and objectives used for their experiments.

Reviewer #2 (Remarks to the Author)

The manuscript by Takaoka and Hamada aims to examine an important question of the contribution of cells to distal visceral endoderm (DVE) during pre implantation development of the mouse embryo. The authors examine embryos at different time points and carry out ablation experiments using a number of transgenic reporter lines. The authors conclude that this distribution is reached at random.

In the opinion of this referee there are several flaws with experiments which have been over-interpreted. Importantly, it is well known already that so called fixed pre-pattern does not exist in the mouse embryo so this is not a novel insight. What is not known is whether in natural, undisturbed development some cells in the mouse embryo are biased to start expressing Lefty and whether this contributes to DVE specification and therefore to laying down the AP axis. The authors' experiments did not address this question. Indeed, removing the Lefty-expressing cells only shows that the rest of the embryo is "regulative" and can substitute for missing cells. It does not address whether these cells were established at random in the first place and what they contribution would have been.

Specific comments:

The quality of many images is very low which often makes it impossible to interpret the data.

The number of embryos examined is also very low (for example, the authors draw some conclusions from as few as 5 embryos).

A major flaw is that the authors do not follow developmental events or trace cells in which they, for example, induced Nodal/Lefty expression throughout implantation. Several papers have been now published using an in vitro system that allows embryos to develop from pre- to post-implantation stages and this system should be used here to address the question authors pose directly.

The authors state "Furthermore, embryos with Nodal mRNA injected into a single blastomere developed a normal A-P axis at E6.5, suggesting that the A-P axis is

established normally even if an ectopic blastomere is chosen to become an L1epi or L1dve cell". However, the authors do not show any evidence that these injected cells show continued upregulation of Nodal - and it is quite possible that they didn't as the amount of growth at implantation is quite dramatic.

Lefty expression in peri-implantation embryos is regulated by Nodal signaling: The authors want to disprove any epiblast origin of DVE cells and to do so they take advantage of the double transgenic lines Lefty-Venus and Oct3-Tomato and demonstrate the absence of a Tomato signal outside epiblast and in Venus+ve cells. I would strongly suggest they use classical immunofluorescence analysis using Oct4 and Lefty antibodies in order to prove no DVE cells can originate from the epiblast.

Figure 1. Here the authors should perform immunofluorescence using Lefty antibodies. They are exploring the different regulative region of Lefty1 but they should confirm the expression of Lefty protein at all the stages they have analyzed. This will give an important support to their message.

In suppl. fig.2a, immunofluorescence to detect Lefty would be helpful.

In suppl. fig.2b: although the authors point out the conservation between mouse and human, they need to give an explanation why the CHIP-seq was performed on human ESCs and not mouse ESCs.

Nodal signaling induces Lefty expression in the targeted or neighboring blastomeres:

The authors should use immunofluorescence for pSmad2 as an additional readout of Nodal activity.

In fig 2b, the authors should add images of A7-venus and lineage markers (at E6.5, Oct4 for Epi and Eomes for VE; at E4.5, Nanos and Gata4,6 or Sox17)

In suppl fig.3 pSmad2 staining is required as additional prove of Nodal/Activin signaling inhibition.

In Fig2 c, the quality of the pictures is poor. Better and clearer pictures should be provided.

The same applies for fig.3a; in particular the quality of the Nodal-Tomato signal is poor. Again, the authors should increase the number of embryo analyzed as it is currently very few.

The authors conclude that the Lefty expression is induced by Nodal-Foxh1 signaling observing Lefty expression after Nodal mRNA injection in one blastomere. This experiment is inconclusive and does not clearly prove what the authors state. It is unclear how a specific blastomere receiving Nodal mRNA should activate Lefty as a result of Nodal-Foxh1 signaling. Moreover, it's not clear in the situation in embryo where a blastomere is injected only with mCherry mRNA. Do the authors observe any variation in number of Lefty+ve cells. This might indicate a real effect of Nodal mRNA injection.

Lefty activity restricts the number of prospective DVE cells:

In suppl. fig. 4a the quality of the images is very poor. The authors should increase the magnification and use a lineage marker in order to clearly demonstrate the lineage positive for LacZ (An anti-lacZ Ab could also be used in IF). To define a lineage just by position would not be considered appropriate when antibodies for specific lineage can be used. The same is true for suppl. fig 4b.

In suppl. fig. 5. IF or FISH should be used to prove the successful knock-out of both Lefty1,2.

In fig.4a. the embryos used as an example look quite different from their developmental stage. The wild type embryo seems to have a smaller epiblast and primitive endoderm and the mutant also looks more advanced because of the onset of parietal endoderm formation.

In line 175, the authors state "Although the number of PROSPECTIVE DVE cells was increased...". I

am afraid I cannot agree until proper lineage tracing is performed from E4.5 to E5.5/E6.5. The authors should culture and image embryos (wild type and mutant) during that period and track the Lefty+ve cells. Such culture and imaging is now possible.

Nodal-Lefty regulatory network: self-enhancement and lateral inhibition (SELI):

In this paragraph the authors confirmed data already published showing enhancers responsible for Nodal expression during pre-implantation. In fig.5c, the authors wish to prove self-enhancement of Nodal - by culturing embryos in Nodal inhibitor and then analyzing Nodal expression. They should check pSmad2 reduction as additional proof of signaling inhibition. The quality of the tomato signal in this image is not very clear. Because the authors have access to several Nodal lines (lacZ and Tomato) in addition to the A7-Venus lines, they should show Nodal reduction using both LacZ and Tomato and as consequence of inhibitor treatment (A7-Venus lines could also be used). The data shown here also contradicts Granier et al 2011. The expression of the ASE-YFP transgene in the blastocyst is unaffected in Nodal-/- or FoxH1-/- embryos, but drastically reduced after treatment with SB-431542. This strengthens the case for the involvement of factors other than Nodal and FoxH1 in Activin/Nodal signaling before implantation). Clarification of this point should be given.

L1epi and L1dve cells are selected randomly in the blastocyst:

In fig.6a: did the authors check if the embryos transferred to mother were effectively injected? If so, how? The authors should analyse an increased number of experimental embryos and control embryos injected with mCherry mRNA only. The quality of the pictures should be improved.

Line 218, I am guessing that the authors do not mean mechanical ablation but laser ablation?

The authors should consider rephrasing lines 218-23. One could interpret their findings if by eliminating Lefty cells, the neighboring cells can begin to respond to Nodal expressing Lefty because the ablation simply removed inhibitory signals. Transferring the embryos back to the mother and then confirming their development confirms the plasticity of the embryo – its ability to recover from the loss of the cells. It proves neither random, de novo or regulated origin nor the predetermination of DVE cells. The remaining conclusions are speculation because the authors have not attempted to perform any live imaging of embryos from E4.0 to E5.5. It is suggested they image and track the newly formed Lefty cells to prove that they will become DVE.

General and final comments:

The aim of the manuscript is valid from the developmental and molecular point of view. The authors have tried to demonstrate that DVE cells are selected in a random and regulated manner. However, the manuscript lacks critical experiments aimed at really proving their hypothesis. For example, live imaging of embryos in the places advised through peri- to post-implantation would have given a direct test of the hypothesis. These issues should be addressed before the manuscript could be considered for publication.

Reviewer #3 (Remarks to the Author)

A & B: Takaoka and Hamada have contributed original research that is relevant to and should be of interest to any mammalian developmental biologist. Their work is a big step in addressing general long standing questions of symmetry breaking and self-organization in the mammalian embryo. Specifically they investigate the first symmetry breaking event of gastrulation, the formation of the A-P axis, and they discover the stage at which this event begins to be determined. Importantly, they also show that determination is a random, but controlled event, involving a Turing activator-inhibitor pair, the activator in this case being Nodal and the inhibitor being Lefty1. Such a mechanism may be a general feature or solution to the problem of breaking symmetry, and thus careful study of this example could be a lesson to the field. The paper is clear,

well-written and the conclusions are new and highly significant. I cannot currently recommended it for publication as I believe there are still points that need to be addressed.

C: The data and methodology in my opinion are valid and well thought out.

D: The researchers used statistics appropriately.

E & F: A large hole I found missing in the paper is that Lefty1,2-/- embryos appear to be normal in development at E6.5 and E7.5, and that ablation of Lefty1+ cells at 3.5 and subsequent 8hr delay until new Lefty expressing cells appeared had no larger delay or effect on development at E6.5 of Lefty. The authors discuss this at some length in their discussion, but do not seem to acknowledge how it weakens their case. Finding an interesting activator-inhibitor patterning mechanism correlated with A-P development is interesting, but if severely perturbing this mechanism does not severely perturb A-P axis formation one cannot help but wonder what is missing and why this mechanism exists at all then. Perhaps there are redundant inhibitors and they need to look at Cerberus as well?

Minor concerns:

- Eomes is misspelled as "Emos" at the beginning of the paper.
- There is an apparent controversy about epi cells contributing to the DVE as shown by ref 12 or by the authors in a previous study. In this context, I do not see why the experiment with the Oct3/4 reporter helps in resolving the debate.
- While the data is convincing, much of it misses clear quantification: the manuscript concludes a reaction/diffusion-like mechanism. Therefore, distances between the sources of secretion for morphogen/inhibitors matter. The authors need to show such quantification for Figure 3 and clearly present the histogram of distances between Nodal and Lefty cells.
- Panel 2c also needs proper quantification as only one image from one embryo is shown.
- I find the schemes of Figure 7c very confusing and not well explained neither within the legend or the main text. The schemes need clarification.
- The manuscript should address the role for Cer in the symmetry breaking process, if any.

Reviewer #4 (Remarks to the Author)

This paper is focused on the issue of how prospective anterior-posterior axis determining cells are selected in the early mouse embryo. The authors focus predominantly on the role of the Nodal antagonist, Lefty1. They show that Lefty1 is first expressed in the ICM in a subset of epiblast progenitor cells and then in a subset of primitive endoderm cells fated to become DVE. The paper uses a combination of gene reporters, mouse mutants and overexpression and cell ablation experiments to investigate how prospective DVE cells are selected. The model that they come up with suggests that Lefty1 expression in the prospective DVE cells is random, but once a cell starts to express Lefty1 (and possibly also Lefty2) it then can inhibit the expression of these antagonists in surrounding cells. The role of these antagonists is thus to restrict the number of DVE cells. The authors conclude that the selection of the prospective DVE cells is thus both random and regulated.

I think that the work is very interesting and novel I think the authors come up with an elegant model showing interplay between Nodal and its antagonists Lefty1 and Lefty2.

I have some general points and some specific ones covering the data themselves the interpretation, additional experiments, presentation of the data and quantification.

1. The quality of the images is generally low and in some cases it is very difficult to see the staining. This is true of Figure 1a, particularly the E4.5 and E6.5 panels for L1-2.0-lacZ. For the E4.5 staining it is not clear where the staining is. Also, some of the fluorescent staining is very faint: Figure 2B, particularly the Cer11 staining, Figure 3a, Figure 6, and Figure 7a and b. In these figures it is really not clear what is going on. They must be improved.

2. It is essential for the results to be convincing that numbers are shown for all the experiments and that it is indicated how many times each experiment was repeated independently.

3. The nomenclature for the different constructs used in Figure 1 is extremely difficult to follow and should be simplified so that it is clear what is in each construct and what is mutated.
4. In Figure 1 the authors investigate the relative roles of the DE and PE. However they do not test a construct that has a DE and a mutated PE. I think this is an important omission.
5. In the text on page 5 the authors state that L1-2.0Fm in which the Foxh1 binding sequences in the PE are mutated was inactive at E4.5. Where are the data to show this?
6. In the same paragraph on page 5 the authors say that the expression level of L1-0.7Fm-LacZ is lower than that of L1-0.7-lacZ. This is not obvious in the images shown.
7. On page 7 the authors describe the experiment where they test the effect of ectopic expression of Nodal on Lefty1 expression. They see it either in the Nodal-expressing cell or in a neighbouring cell. Why the difference? Why do not all Nodal-expressing cells express Lefty1 as well? This needs to be investigated as it may be important for the overall model.
8. The really unexpected result is that ectopic Nodal induction generates ectopic L1^{epi} cells, but has no effect on embryo development. Similarly when the L1^{dve} or L1^{epi} cells are ablated, this has no effect on patterning. This needs to be explored more thoroughly. It would appear that the system is extremely well buffered, but this needs to be proven. What happens if one of the Leftys is overexpressed in a single blastomere at E3.2?
9. In the first paragraph of the discussion the authors point out that in the Lefty1,2^{-/-} mutant more DVE cells are produced, but even though the DVE is known to guide the migration of the AVE there are no effects on AVE in these embryos. They need to be able to explain why this is the case.

Response to the Reviewers' comments:

Reviewer #1:

In this manuscript Takaoka and Hamada propose that future distal visceral endoderm cells (DVE) are already selected at blastocyst stages within the primitive endoderm (PrE).

Using live and static imaging, as well as some rather elaborate transgenic studies, the authors examine the role and regulation of Lefty1 in early DVE specification. They show that Lefty1 is expressed in a subset of epiblast cells (L1epi) at E3.5 and later in a subset of PrE cells (L1DVE) at E4.5.

The authors also show that this Lefty1 expression is dependent on Nodal-Foxh1. Lefty1 expression is regulated by two enhancers, designated as distal (DE) and proximal enhancer (PE) both containing FoxH1 binding sites. Analysis of lacZ reporter expression of a variety of Lefty1 transgenes containing both (DE and PE) enhancers, only the PE or mutated DE or PE, respectively, leads the authors to conclude that the PE drives Lefty1 expression in prospective DVE cells, and the DE in epiblast cells.

Further evidence for a Nodal dependent Lefty1 expression was obtained by injection of Nodal mRNA into (individual?) ICM cells induced Lefty1 expression, whereas inhibition of Nodal signaling prevented Lefty1 expression and led to an increase in DVE cells, as did the deletion of Lefty1, and Lefty1 and Lefty2. However neither of these scenarios affected anterior-posterior (A-P) axis formation which is a functional consequence of the migration of the DVE. A-P axis establishment was also apparently unaffected through the ablation of L1DVE cells blastocyst stage embryos.

Collectively these data lead the authors to posit that no fixed pre-pattern of the future A-P axis exists in the blastocyst, and that the selection of L1DVE cells within any given embryos is random and regulated by Nodal activity. These are important open questions that are difficult to address, but which lie at the heart of the regulative nature of the mammalian embryo. However, in this reviewer's opinion the conclusions drawn, and the proposed model are not well supported by the data which are generally not rigorously analyzed.

General comments:

- The resolution of the analysis presented in this study needs to be improved. High magnification views of ICMs should be shown in the figures.*

Response: We now show most of the images at a higher magnification with a higher resolution

- *Blastocysts stages should be classified by cell numbers rather than by embryonic (E) day.*

Response: We now indicate the number of cells for many of pre-implantation embryos (Fig. 1; Fig. 2, Fig. 3, Fig.4, Fig.5, Fig. 6, Fig.7, Supplementary Fig. 2, Supplementary Fig. 4).

- *Throughout the paper LacZ expression in blastocysts is hardly visible in the chromogenic stains, and should be rather analyzed with anti-LacZ antibodies which provide single cell resolution. Blastocysts should be counterstained with DAPI or co-stained with ICM and PrE specific markers for improved resolution of analysis.*

Response: For most of the experiments that involved LacZ staining in blastocysts, we have replaced LacZ transgenes by Venus transgenes, and have re-done experiments. Blastocysts have been counterstained with DAPI, and GATA6 (Fig. 1, Fig.2, Supplementary Fig. 2, Supplementary Fig. 4).

- *In many instances resolution of data presented does not allow one to distinguish between various scenarios and outcomes.*

Response: We have improved resolution of images.

- *Data should be analyzed more quantitatively.*

Response: Yes, we have performed quantitative analysis such as by counting cell numbers.

Major Comments:

Figure 1:

a: lacZ staining in embryos is hardly visible.

The localization of the reporter (using antibodies against lacZ or Venus) should be shown using fluorescence, so that embryos can be co-stained with ICM (Nanog) and PrE (Gata6) specific markers.

Response: To locate cells more clearly, we have replaced LacZ reporters by Venus reporters. Embryos are counter-stained with DAPI and co-stained with Gata6, which allows the localization of Venus⁺ cells clearly in a blastocyst.

b: Authors should put embryos expressing the various transgenic constructs into culture in the presence of inhibitors of Nodal signaling and vs. Nodal, and determine the effects on the population within the ICM.

Response: We have examined the effects of SB431542 (an inhibitor of Nodal signaling) on Lefty1(Cherry) BAC and A₇Venus (Fig.2), L1-DE⁺PE⁺mVenus (Supplementary Fig. S2h) and Lefty1(mVenus) BAC (Supplementary Fig. S2h), and have confirmed that their expression depends on Nodal signaling.

Figure 2:

a: *lacZ* should be analyzed by IF. *Smad2* expression is not distinguishable from background stain in this panel. Authors should use a p*Smad2* antibody.

Response: We tested several *Smad2* and p*Smad2* antibodies, but none of them worked for staining of blastocysts (shown bellow). Therefore, We have examined *Smad2* expression by a *Smad2*(Venus) transgene. New results (Fig. 2a) show that *Smad2* is expressed in all blastomeres of a blastocyst.

p*Smad2* antibodies, which is known to work well with E5.5 and E8.5 embryos, did not work with blastocysts. In spite of Nodal signaling activation and inhibition, the staining pattern (light blue) did not change. Nuclei were indicated in blue. Scale bar is 50um.

b: Authors should show high magnification views of these stages as well as intermediate stages and merges of the A₇-Venus and *Cer1* and *Lefty1,2*, respectively.

Response: We now show higher magnification views and merges of the A₇-Venus and Cer1 and Lefty1,2.

c: Single-plane of section, not rendering should be shown for the brightfield channel. It would be interesting to see the expression of A₇-Venus and Lefty1-Cherry in the presence of Nodal vs Nodal inhibitors.

Response: We now provide single-plane of sections. We have examined more samples, 16 samples in total. A₇-Venus and Lefty1(Cherry) were co-expressed in 15/16 embryos. Addition of SB431542 abolished A₇-Venus and Lefty1(Cherry) expression.

Figure 3:

b: Stage of blastocyst needs to be defined better.

Response: We counted the cell number of injected embryos after immunostaining (it is impossible to count cell number before injection). The cell numbers are indicated in the figure.

c: If overlaid on the brightfield channel the cherry positive cell is not ICM! Authors should show merge of DAPI channel.

Response: We now provide DAPI images and their merges with other markers.

c and d: Authors should include DAPI channel to show all cells and distinguish who is a neighbor, and who is not, and include histograms of cell numbers.

Response: We now provide DAPI images and merges, so that all cells can be seen. To identify the injected cells more clearly, mRNA for memTomato (membrane-localized Tomato) was injected instead of mRNA for Cherry. This has made it easier to know spatial localization of injected cells and neighboring cells. We also provide histograms showing the number of cells.

Figure 4:

a: Authors should include the cell number for each stage, otherwise it is very difficult to compare between stages.

Response: We now provide the cell number for each embryo. The WT embryo is replaced by a new embryo whose cell number matches with that of other embryos.

b: Authors should include some statistics on the data, since if it weren't for one data point at 50 in Lefty1,2^{-/-}, Lefty1^{-/-} and Lefty1,2^{-/-} may be the same. In addition, can the authors comment on if the total number of Gata6⁺ cells were increased or the same. Perhaps data should be plotted as box plot?

Response: As suggested, we have counted the number of Gata6⁺ cells, which did not change between WT embryo and Lefty mutant embryos. We also counted the number of Lefty1⁺ cells in each genotype, and results are shown in quantitative histograms as the box plots. It is clear that the number of Lefty1⁺ cells increases in the Lefty1,2(-/-) embryo. We also counted the number of Oct3/4⁺ cells in each embryo. These are now in Fig. 4b-e.

Figure 5:

b: Numbers should be provided in this section for: embryos analyzed, embryos showing a defect, cells in embryos that are affected. Furthermore better quality images should be used, reviewer could not make out what is going on in the images of E4.5 embryos.

Response: We now provide the number of embryos examined, the number of embryos showing a defect.

c: this reviewer is not sure if the tomato expression corresponds with the localization of the ICM in this panel.

In addition, what happens if embryos are cultured in Nodal?

Response: Embryos with Nodal (Tomato) BAC transgene are now stained with Gata6 and DAPI. Tomato⁺ cells (Nodal-expressing cells) are located in the ICM. We have examined the effect of SB43142, and have found that Nodal expression is down-regulated by SB43142.

d: Could authors include earlier stages and give numbers of embryos analyzed?

Response: Unfortunately, we were unable to examine earlier expression, because we could not obtain Nodal (lacZ)BAC;Foxh1(-/-) embryos until this revision.

Figure 6:

The ablation experiments in theory are nice, but the authors do not show any controls for these studies.

How can the authors tell that they ablated and not bleached an mVenus cell?

Cell death markers to distinguish bleaching vs. ablating should be included.

Numbers of cells in ICM before and after ablation (to confirm cell ablation) should be included.

How do embryos look like immediately after ablation? Are any mVenus + cells left in the ICM after ablation?

Ablation experiments should be better explained in the text or methods. Authors should state in the text that these are laser ablations, not manual ablations!

Authors should clarify which laser or scope they used for the ablation. In the methods section they state the use of a two- or multi-photon scope and the

488nm laser, however, this laser line is not multi-photon, but a standard confocal line. A two-photon laser would be an IR laser which for ablating Venus or GFP would be in the 800 or 900nm range.

Authors should clarify if cell are ablated in only a single z-plane or in a total z-stack, hitting more than one cell.

Response: By employing membrane-anchored Venus (mVenus), laser was focused at the center of cell, so that cells would be ablated not bleached. To confirm cell ablation, we monitored changes in cell membrane of target mVenus⁺ cells. When laser ablation was successful, we see cell debris after laser ablation (Fig. 6c). All mVenus⁺ cells were ablated, so that there was no mVenus⁺ cell left after the ablation. We used laser ablations (not manual ablations), and details of ablation experiments are now described in Methods.

Figure 7:

a and b should go into figure 6 and as panels presented in figure 6 need to be better control for.

*Response: Yes, it would be better to show **a** and **b** in Figure 6, but there was not enough space to place all the pictures (Fig. 6 and Fig7a,b) in a single figure.*

c: Authors have assessed Nodal expression, not the activity, if they want to discuss the activity they need to use the FoxH1 A7- transgene or pSmad2 localization.

Response: We now describe as Nodal expression instead of Nodal activity.

Minor points:

Overall, the manuscript should be checked for typos and wording.

Movies should be edited to draw the reader/viewer's attention to specific details.

Introduction:

Line 44-46: sentence does not make sense. "We know how the A-P axis is established at the level of the DVE. The questions should be how DVE is specified."

Line 46-48: sentence contradicts previous sentence.

Response: We have improved the sentence accordingly.

Material and methods:

Time lapse microscopy and image processing:

Authors should include information on z-intervals and objectives used for their experiments.

Response: We now provide information on z-intervals and objectives in Materials and Methods (page 13).

Reviewer #2 (Remarks to the Author):

The manuscript by Takaoka and Hamada aims to examine an important question of the contribution of cells to distal visceral endoderm (DVE) during pre implantation development of the mouse embryo. The authors examine embryos at different time points and carry out ablation experiments using a number of transgenic reporter lines. The authors conclude that this distribution is reached at random.

In the opinion of this referee there are several flaws with experiments which have been over-interpreted. Importantly, it is well known already that so called fixed pre-pattern does not exist in the mouse embryo so this is not a novel insight. What is not known is whether in natural, undisturbed development some cells in the mouse embryo are biased to start expressing Lefty and whether this contributes to DVE specification and therefore to laying down the AP axis. The authors' experiments did not address this question. Indeed, removing the Lefty-expressing cells only shows that the rest of the embryo is "regulative" and can substitute for missing cells. It does not address whether these cells were established at random in the first place and what their contribution would have been.

Specific comments:

The quality of many images is very low which often makes it impossible to interpret the data.

Response: We have improved the quality of these images.

The number of embryos examined is also very low (for example, the authors draw some conclusions from as few as 5 embryos).

Response: We have increased the number of samples examined for Fig. 3, Fig. 4. It is clear from quantitative analysis (Fig. 4b-e) that Lefty1^{dve} cells increase in Lefty mutants.

A major flaw is that the authors do not follow developmental events or trace cells in which they, for example, induced Nodal/Lefty expression throughout implantation. Several papers have been now published using an in vitro system

that allows embryos to develop from pre- to post-implantation stages and this system should be used here to address the question authors pose directly.

Response: We agree that it would be nicer if one can follow the fate of Lefty1^{dve} cells by live imaging. We have attempted to establish in vitro culture method, in which pre-implantation embryos can develop until gastrulation stage (Morris et al., 2012; Bedzhov et al, 2014). We followed the same protocol and have tried several times, but in our hand, the frequency of successfully developed embryo was very low (1/20 embryos when E3.5 pre-implantation stage embryos were cultured; 0/23 embryos when E4.5 peri-implantation stage embryos were cultured): in most of the cases, embryos failed to develop into a 3-D structure like an in vivo embryo and lost Lefty1 expression (shown below). Unfortunately, we are unable to follow the Lefty1^{dve} cells in live imaging.

E4.5 embryos with L1-PE⁺ Venus were cultured in vitro as described by others (Morris et al., 2012; Bedzhov et al, 2014). Note that morphology of the embryo is abnormal at 50h and 118 hours and that Lefty1 expression is lost quickly.

The authors state “Furthermore, embryos with Nodal mRNA injected into a single blastomere developed a normal A-P axis at E6.5, suggesting that the A-P axis is established normally even if an ectopic blastomere is chosen to become an L1epi or L1dve cell”. However, the authors do not show any evidence that these injected cells show continued upregulation of Nodal - and it is quite possible that they didn’t as the amount of growth at implantation is quite dramatic.

Response: To examine the fate of an injected blastomere, E3.2 embryos harboring Lefty1(mVenus) BAC and R26R were injected with Nodal mRNA, Cre mRNA and mCherry mRNA. Injected embryos with a mCherry⁺ cell were transferred to mother, recovered at E6.5, examined for mVenus fluorescence and were stained for LacZ (Fig. 6d). LacZ⁺ cells were found in DVE-derived cells, suggesting that an injected cell survived, proliferated and contributed to DVE. Venus⁺ cells (Lefty1⁺ cells) were found in DVE-derived cells and AVE, which is the normal Lefty1 expression pattern. Nodal protein is produced in the injected

cell (Fig. 3h), and induces Lefty1 expression in the same/neighbor cell (Fig. 3c). As suggested by the reviewer, the Nodal protein produced would be gradually diluted upon cell division. Therefore, Lefty1^{dve} cells may not need continuous high level of Nodal until they become DVE cells.

Lefty expression in peri-implantation embryos is regulated by Nodal signaling: The authors want to disprove any epiblast origin of DVE cells and to do so they take advantage of the double transgenic lines Lefty-Venus and Oct3-Tomato and demonstrate the absence of a Tomato signal outside epiblast and in Venus+ve cells. I would strongly suggest they use classical immunofluorescence analysis using Oct4 and Lefty antibodies in order to prove no DVE cells can originate from the epiblast.

Response: We now use antibodies to Lefty, Oct3/4 and Sox2. Cells positive for Sox2 and Oct3/4 were never found in the DVE at E5.5 and DVE and AVE at E6.0 (Supplementary Fig. 1).

Figure 1. Here the authors should perform immunofluorescence using Lefty antibodies. They are exploring the different regulative region of Lefty1 but they should confirm the expression of Lefty protein at all the stages they have analyzed. This will give an important support to their message.

Response: We agree that it is important to examine endogenous Lefty1 protein. We have tested anti-Lefty antibodies that recognize both Lefty1 and Lefty2. Although they can detect Lefty proteins in embryos older than E5.5 (Supplementary Fig. 1b-d, Fig. 2b, 6d, 7b), they cannot do so with earlier embryos (E3.5~E4.5). This may be due to a lower level of Lefty protein or to unsuitable fixation condition for earlier embryos. However, various BAC and plasmid reporter transgenes labeled the same cells in a blastocyst: Lefty1(Cherry) BAC, L1-0.7(L1-DE⁺PE⁺)-mVenus, L1-PE⁺-mVenus (Supplementary Fig. 2b, 2c), suggesting that these cells do express endogenous Lefty1 gene.

In suppl. fig.2a, immunofluorescence to detect Lefty would be helpful.

In suppl. fig.2b: although the authors point out the conservation between mouse and human, they need to give an explanation why the ChIP-seq was performed on human ESCs and not mouse ESCs.

Response: Unfortunately, Lefty antibodies cannot detect endogenous Lefty proteins in pre-implantation embryos, as mentioned above. The ChIP-seq data were available from human ESCs but not from mouse ESCs, which is the reason why we used the data from human ESCs.

Nodal signaling induces Lefty expression in the targeted or neighboring blastomeres: The authors should use immunofluorescence for pSmad2 as an

additional readout of Nodal activity. In fig 2b, the authors should add images of A7-venus and lineage markers (at E6.5, Oct4 for Epi and Eomes for VE; at E4.5, Nanos and Gata4,6 or Sox17)

Response: The antibody against pSmad2 works out for E5.5~E6.5 embryos, but somehow does not work out with E3.5~E4.5 embryos. We believe that A₇Venus serves as a suitable readout of Nodal activity. As suggested by the reviewer, we now use Gata6 at E3.5; Oct3/4 at E6.0. in Fig. 2b.

In suppl fig.3 pSmad2 staining is required as additional prove of Nodal/Activin signaling inhibition.

Response: As mentioned above, the antibody against pSmad2 does not work out with E3.5~E4.5 embryos (the signal/noise ratio is low). A₇-hsp cassette, which is known to measure Nodal signaling activity in gastrulating mouse embryos, Xenopus embryo explants and in cultured cells (Saijoh et al., 2000), is the best reporter that can monitor Nodal activity. A₇-Venus expression was lost by 10 μM SB431542 (Supplementary Fig. 3a), indicating that Nodal/Activin signaling is inhibited by 10 μM SB431542.

In Fig2 c, the quality of the pictures is poor. Better and clearer pictures should be provided. The same applies for fig.3a; in particular the quality of the Nodal-Tomato signal is poor. Again, the authors should increase the number of embryo analyzed as it is currently very few.

Response: Pictures in Fig. 2c and Fig. 3a have been improved and they are now clear. We have increased the number of samples. Among 24 embryos examined, Lefty1 was induced in the same cells that had earlier initiated Nodal expression (12/24 embryos) or in their neighboring cells (11/24 embryos).

The authors conclude that the Lefty expression is induced by Nodal-Foxh1 signaling observing Lefty expression after Nodal mRNA injection in one blastomere. This experiment is inconclusive and does not clearly prove what the authors state. It is unclear how a specific blastomere receiving Nodal mRNA should activate Lefty as a result of Nodal-Foxh1 signaling. Moreover, it's not clear in the situation in embryo where a blastomere is injected only with mCherry mRNA. Do the authors observe any variation in number of Lefty+ve cells. This might indicate a real effect of Nodal mRNA injection.

Response: We have counted the number of Lefty^{dve} cells at E4.5 in the injected embryos (Fig. 3e). There was no significant difference between Nodal plus Cherry -injected and Cherry alone-injected embryos (five Lefty^{dve} cells in both cases at E4.5).

Lefty activity restricts the number of prospective DVE cells:

In suppl. fig. 4a the quality of the images is very poor. The authors should increase the magnification and use an lineage marker in order to clearly demonstrate the lineage positive for LacZ (An anti-lacZ Ab could also be used in IF). To define a lineage just by position would not be considered appropriate when antibodies for specific lineage can be used. The same is true for suppl. fig 4b.

Response: As suggested, we now employ Venus and mTomato in addition to LacZ, and use a lineage marker (Gata6) (Supplementary Fig. 4b). These new data confirm that Lefty1 is expressed in a group of PrE cells but is not expressed in EPI. Lefty2 is expressed in a group of EPI and in a subpopulation of Lefty1⁺ PrE cells. Pictures are shown at a larger magnification.

In suppl. fig. 5. IF or FISH should be used to prove the successful knock-out of both Lefty1,2.

Response: We now show by IF that Lefty1, 2 proteins are absent in Lefty1,2 double mutant embryos (Supplementary Fig. 5b).

In fig.4a. the embryos used as an example look quite different from their developmental stage. The wild type embryo seems to have a smaller epiblast and primitive endoderm and the mutant also looks more advanced because of the onset of parietal endoderm formation.

Response: We have counted the number of EPI and PrE cells in each embryo. The quantitative data are now shown in Fig. 4b-e. The reviewer is correct that the WT embryo shown in Fig. 4a was less advanced. This WT embryo is now replaced by another WT embryo that has comparable number of cells.

In line 175, the authors state “Although the number of PROSPECTIVE DVE cells was increased...”. I am afraid I cannot agree until proper lineage tracing is performed from E4.5 to E5.5/E6.5. The authors should culture and image embryos (wild type and mutant) during that period and track the Lefty+ve cells. Such culture and imaging is now possible.

Response: Our previous data obtained by genetic fate mapping have shown that Lefty1^{dve} cells at E3.5~4.2 contribute to DVE cells at E5.5. We fully agree that it would be nicer if one can follow the fate of Lefty1^{dve} cells by live imaging. As mentioned above, we have attempted to establish in vitro culture method, in which pre-implantation embryos can develop until gastrulation stage (Morris et al., 2012; Bedzhov et al, 2014). We have followed the same protocol and have tried very hard, but in our hand, the frequency of successful embryo was very low (1/43 embryos): in the remaining cases, embryos failed to develop into a 3-D structure. Therefore, we are unable to follow the Lefty^{dve} cells in live imaging.

Nodal-Lefty regulatory network: self-enhancement and lateral inhibition (SELI):

In this paragraph the authors confirmed data already published showing enhancers responsible for Nodal expression during pre-implantation. In fig.5c, the authors wish to prove self-enhancement of Nodal - by culturing embryos in Nodal inhibitor and then analyzing Nodal expression. They should check pSmad2 reduction as additional proof of signaling inhibition. The quality of the tomato signal in this image is not very clear. Because the authors have access to several Nodal lines (lacZ and Tomato) in addition to the A7-Venus lines, they should show Nodal reduction using both LacZ and Tomato and as consequence of inhibitor treatment (A7-Venus lines could also be used). The data shown here also contradicts Granier et al 2011. The expression of the ASE-YFP transgene in the blastocyst is unaffected in Nodal^{-/-} or FoxH1^{-/-} embryos, but drastically reduced after treatment with SB-431542. This strengthens the case for the involvement of factors other than Nodal and FoxH1 in Activin/Nodal signaling before implantation). Clarification of this point should be given.

Response: As suggested by the reviewer, we have examined Nodal expression with Nodal (tomato) BAC transgene and show that Nodal expression is down-regulated by 10 μ M SB431542 (Fig. 5c). Furthermore, Nodal (lacZ) BAC expression was down-regulated in Foxh1(-/-) embryos. As indicated by the reviewer, this may contradict the Granier et al (2011) paper, where expression of ASE-YFP transgene was unaffected in Nodal (-/-) and Foxh1(-/-) embryos but is down-regulated by 40 μ M SB431542 at E4.5. Several possibilities may underlie this potential contradiction. First, ASE-YFP transgene is composed of a ~300 bp region that acts as a Nodal-responsive enhancer, whereas our Nodal reporters are all BAC transgenes. Although ASE-YFP seems to recapitulate Nodal expression at E5.5, it may not do so at different stages. It is generally true that BAC transgenes recapitulate endogenous gene expression more faithfully than short transgenes do. Curiously, ASE-YFP marked the epiblast, whereas our A7-Venus (Nodal-Foxh1-dependent reporter) is active in a subpopulation of the primitive endoderm (Figure 2b) at E4.5. Secondly, the concentration of SB431542 is different between two studies: 10 μ M in our study, 40 μ M in their study. Our data clearly show that 10 μ M is sufficient to inhibit Nodal signaling (also Yamamoto et al., 2009). Much higher concentration (40 μ M) could have non-specific effects on embryo development.

L1epi and L1dve cells are selected randomly in the blastocyst:

In fig.6a: did the authors check if the embryos transferred to mother were effectively injected? If so, how? The authors should analyse an increased number of experimental embryos and control embryos injected with mCherry mRNA only. The quality of the pictures should be improved. Line 218, I am guessing that the authors do not mean mechanical ablation but laser ablation?

Response: We injected mCherry mRNA with or without Nodal mRNA. Injected embryos were examined for mCherry expression, and only embryos with a mCherry⁺ cell were transferred to mother. In some cases (Fig. 6b), embryos with Lefty1(mVenus) BAC and R26R were injected with Nodal mRNA, Cre mRNA and mCherry mRNA. Injected embryos were transferred to mother, recovered at E6.5, examined for mVenus fluorescence and were stained for LacZ. In 11/11 embryos examined, LacZ⁺ cells were found in DVE-derived cells, suggesting that an injected cell survived, proliferated and contributed to DVE. Venus⁺ cells (Lefty1⁺ cells) were found in DVE-derived cells and AVE (11/11 embryos), which is the normal Lefty1 expression pattern. As pointed out by the reviewer, we did not mean mechanical ablation but laser ablation (the methods for laser ablation is now described in Experimental Procedures).

The authors should consider rephrasing lines 218-23. One could interpret their findings if by eliminating Lefty cells, the neighboring cells can begin to respond to Nodal expressing Lefty because the ablation simply removed inhibitory signals. Transferring the embryos back to the mother and then confirming their development confirms the plasticity of the embryo – its ability to recover from the loss of the cells. It proves neither random, de novo or regulated origin nor the predetermination of DVE cells. The remaining conclusions are speculation because the authors have not attempted to perform any live imaging of embryos from E4.0 to E5.5. It is suggested they image and track the newly formed Lefty cells to prove that they will become DVE.

Response: We have improved lines 218-223, according to the reviewer's suggestion. It would be nice if can follow the behaviors of L1^{dve} cells by live imaging, but as mentioned above, we were unable to establish efficient embryo culture system from E4.0 to E5.5 despite of rigorous attempts. We feel that this will be the next issue.

General and final comments:

The aim of the manuscript is valid from the developmental and molecular point of view. The authors have tried to demonstrate that DVE cells are selected in a random and regulated manner. However, the manuscript lacks critical experiments aimed at really proving their hypothesis. For example, live imaging of embryos in the places advised through peri- to post-implantation would have given a direct test of the hypothesis. These issues should be addressed before the manuscript could be considered for publication.

Reviewer #3:

A & B: Takaoka and Hamada have contributed original research that is relevant to and should be of interest to any mammalian developmental biologist. Their work is a big step in addressing general long standing questions of symmetry

breaking and self-organization in the mammalian embryo. Specifically they investigate the first symmetry breaking event of gastrulation, the formation of the A-P axis, and they discover the stage at which this event begins to be determined. Importantly, they also show that determination is a random, but controlled event, involving a Turing activator-inhibitor pair, the activator in this case being Nodal and the inhibitor being Lefty1. Such a mechanism may be a general feature or solution to the problem of breaking symmetry, and thus careful study of this example could be a lesson to the field. The paper is clear, well-written and the conclusions are new and highly significant. I cannot currently recommended it for publication as I believe there are still points that need to be addressed.

C: The data and methodology in my opinion are valid and well thought out.

D: The researchers used statistics appropriately.

E & F: A large hole I found missing in the paper is that Lefty1,2^{-/-} embryos appear to be normal in development at E6.5 and E7.5, and that ablation of Lefty1⁺ cells at 3.5 and subsequent 8hr delay until new Lefty expressing cells appeared had no larger delay or effect on development at E6.5 of Lefty. The authors discuss this at some length in their discussion, but do not seem to acknowledge how it weakens their case. Finding an interesting activator-inhibitor patterning mechanism correlated with A-P development is interesting, but if severely perturbing this mechanism does not severely perturb A-P axis formation one cannot help but wonder what is missing and why this mechanism exists at all then. Perhaps there are redundant inhibitors and they need to look at Cerberus as well?

Minor concerns:

- Eomes is misspelled as “Emos” at the beginning of the paper.

Response: This is now corrected.

- There is an apparent controversy about epi cells contributing to the DVE as shown by ref 12 or by the authors in a previous study. In this context, I do not see why the experiment with the Oct3/4 reporter helps in resolving the debate.

Response: we now use Oct3/4 (reporter and antibody) and Sox2 as markers for the epiblast. We show that Lefty1⁺ cells at E5.5 (DVE cells) are all negative for Oct3/4 and negative for Sox2 (Supplementary Fig. 1), suggesting that all DVE cells are derived from Lefty1^{dev} cells, as we previously described (page 4).

- While the data is convincing, much of it misses clear quantification: the manuscript concludes a reaction/diffusion-like mechanism. Therefore, distances between the sources of secretion for morphogen/inhibitors matter. The authors need to show such quantification for Figure 3 and clearly present the histogram of distances between Nodal and Lefty cells.

Response: Lefty1 expression was induced in Nodal-expressing cells (therefore, no distance between both)(12/24 embryos examined) or in cells adjacent to the Nodal-expressing cells (11/24 embryos). This is described on page 6.

- Panel 2c also needs proper quantification as only one image from one embryo is shown.

Response: We have examined 16 embryos in total. In 15/16 embryos examined, Lefty1 expression (Cherry) was found in the same cell positive for A₇Venus. This is now mentioned on page 6.

- I find the schemes of Figure 7c very confusing and not well explained neither within the legend or the main text. The schemes need clarification.

Response: we now explain better in the legend.

- The manuscript should address the role for Cer in the symmetry breaking process, if any.

Response: As requested by the reviewer, we have examined the role of Cer1 in A-P axis formation. In triple mutant embryo lacking Lefty1, Lefty2 and Cer1, Hex⁺ cells (DVE/AVE cells) expanded wider when compared to Lefty1,2 double mutant embryo (Supplementary Fig. 5e). Therefore, Cer1 has a redundant role. This is now mentioned on page 9.

Reviewer #4:

This paper is focused on the issue of how prospective anterior-posterior axis determining cells are selected in the early mouse embryo. The authors focus predominantly on the role of the Nodal antagonist, Lefty1. They show that Lefty1 is first expressed in the ICM in a subset of epiblast progenitor cells and then in a subset of primitive endoderm cells fated to become DVE. The paper uses a combination of gene reporters, mouse mutants and overexpression and cell ablation experiments to investigate how prospective DVE cells are selected. The model that they come up with suggests that Lefty1 expression in the prospective DVE cells is random, but once a cell starts to express Lefty1 (and possibly also Lefty2) it then can inhibit the expression of these antagonists in surrounding cells. The role of these antagonists is thus to restrict the number of DVE cells. The authors conclude that the selection of the prospective DVE cells is thus both random and regulated.

I think that the work is very interesting and novel I think the authors come up with an elegant model showing interplay between Nodal and its antagonists Lefty1 and Lefty2.

I have some general points and some specific ones covering the data themselves the interpretation, additional experiments, presentation of the data and quantification.

1. The quality of the images is generally low and in some cases it is very difficult to see the staining. This is true of Figure 1a, particularly the E4.5 and E6.5 panels for L1-2.0-lacZ. For the E4.5 staining it is not clear where the staining is. Also, some of the fluorescent staining is very faint: Figure 2B, particularly the Cerl1 staining, Figure 3a, Figure 6, and Figure 7a and b. In these figures it is really not clear what is going on. They must be improved.

Response: As described above, we have improved most of the images either by replacing LacZ by Venus or by showing pictures at a larger magnification or at a higher resolution.

2. It is essential for the results to be convincing that numbers are shown for all the experiments and that it is indicated how many times each experiment was repeated independently.

Response: We have increased the number of the embryos examined in most of the experiments, and the numbers of embryos examined are indicated in the text and legend.

3. The nomenclature for the different constructs used in Figure 1 is extremely difficult to follow and should be simplified so that it is clear what is in each construct and what is mutated.

Response: We have changed the nomenclature for the different constructs in a simpler way so that readers can guess what is included in each construct: for example, L1-0.7----→L1-DE⁺PE⁺.

4. In Figure 1 the authors investigate the relative roles of the DE and PE. However they do not test a construct that has a DE and a mutated PE. I think this is an important omission.

Response: We have tested such a construct, L1-DE⁺(LacZ or Venus). This construct was active in ICM at E3.5, and exhibited only ectopic expression in epiblast between E4.5 and E6.5 (Fig. 1b, Supplementary Figure 2e). Therefore, DE is active in ICM at E3.5 but is inactive in PrE at E4.5.

5. In the text on page 5 the authors state that L1-2.0Fm in which the Foxh1 binding sequences in the PE are mutated was inactive at E4.5. Where are the data to show this?

Response: This is included in Fig. 1b (L1-PE^m).

6. In the same paragraph on page 5 the authors say that the expression level of L1-0.7Fm-LacZ is lower than that of L1-0.7 -lacZ. This is not obvious in the images shown.

Response: Staining time was different: 15 minutes for L1-0.7 (L1-DE⁺PE⁺lacZ; Fig. 1c), 12hours for L1-0.7Fm(L1-DE^mPE⁺lacZ; Supplementary Fig. 2d). This is now mentioned on page 5.

7. On page 7 the authors describe the experiment where they test the effect of ectopic expression of Nodal on Lefty1 expression. They see it either in the Nodal-expressing cell or in a neighbouring cell. Why the difference? Why do not all Nodal-expressing cells express Lefty1 as well? This needs to be investigated as it may be important for the overall model.

Response: Since *Nodal* encodes a secreted protein, Nodal protein produced from the injected mRNA would act on the injected cell or its nearby cell, and induce Lefty1 expression. Lefty1 protein produced from the earliest Lefty1-expressing cell (either Nodal-injected cell or neighboring cell) would repress Nodal signaling (and Lefty1 expression) in neighboring cells, restricting Lefty1 expression to a single cell. We believe that this is a reasonable speculation, but is difficult to further test experimentally.

8. The really unexpected result is that ectopic Nodal induction generates ectopic L1^{epi} cells, but has no effect on embryo development. Similarly when the L1^{dve} or L1^{epi} cells are ablated, this has no effect on patterning. This needs to be explored more thoroughly. It would appear that the system is extremely well buffered, but this needs to be proven. What happens if one of the Leftys is overexpressed in a single blastomere at E3.2?

Response: We had performed such experiments before. Injection of Lefty1 mRNA into a single blastomere at E3.2 did not have profound effects on the number of Lefty1⁺ DVE cells at E5.5. Our interpretation is that by the time when Lefty1 protein is produced and secreted from the injected cell, L1^{dve} and L1^{epi} cells already exist, and ectopic Lefty1 protein would not persist until E4.5 when L1^{dve} cells are established via PE.

9. In the first paragraph of the discussion the authors point out that in the Lefty1,2^{-/-} mutant more DVE cells are produced, but even though the DVE is known to guide the migration of the AVE there are no effects on AVE in these embryos. They need to be able to explain why this is the case.

Response: From our previous report (Takaoka et al., NCB 2010), we believe that the role of DVE is to guide AVE migration toward the future anterior side. Even though more DVE cells are produced in Lefty1, 2^(-/-) embryos, they are not dispersed but located on one side (the same side). Therefore, a larger number of

DVE cells would not impair AVE migration. In *Lefty1,2(-/-),Cer1(-/-)* triple mutant embryos (1/2 embryos examined), however, AVE was located normally but was expanded at E6.5 (Supplementary Fig.5e). We speculate that more VE cells are recruited to become AVE cells due to increased migration of DVE cells. This is briefly mentioned in the text (page 8).

Reviewers' Comments:

Reviewer #1:

Remarks to the Author:

Takaoka et al. have substantially improved their manuscript, and have addressed most of the reviewer's comments to satisfaction.

However, some minor points still need to be addressed before the manuscript can be considered for publication.

Figure 3:

Authors state in text that *lefty1* expression began either in the blastomere that received Nodal mRNA (6/19 embryos) or in a neighboring cell (12/19 embryos).

6+12=19 (100%). In Fig. 3g, however, authors state that 5% of embryos show *Lefty1* expression in more distant cells. Numbers here do not add up.

Fig. 3d in this reviewer's mind shows rather the expression of *Lefty1*(mVenus) in a distant cell than as the authors claim in a neighboring cell.

Also, the DAPI channel in this figure is overexposed so it is hard to make out single cells.

The manuscript still contains a lot of typos in the text and figure legends.

Some time stamps in legends of figure 3 and 4 differ from the one's in the actual figures, e.g. Fig 2 c and d.

Reviewer #2:

Remarks to the Author:

In the revised manuscript Takaoka and Hamada have tried to address most of the criticisms raised by the referee but some points remain not explored and not well explained. The authors should address both minor and major points that haven't been addressed.

Minor points and comments:

The following points are considered minor and they are requested to improve the quality of the messages of the manuscript

1) Quality of the pictures: the authors have successfully improved the quality of the pictures nevertheless some further improvement would be required to clearly present the data. In general, the following panels seem to have strong artificial enhancement of the signal (very high contrast and brightness) and lack of suitable cellular resolution (that should be easily achieved using a confocal microscope as mentioned in the Methods in addition to presenting the DAPI signal to reveal the nucleus where not shown).

Fig. 2a,c; Fig.3h (this image is not clear – Nodal protein and Cherry appear merged and the cellular resolution is of poor quality).

Fig. 7b strong contrast seems to have been used to hide the background/noise relative to the staining for *Lefty* and *Cer11*.

Fig. 6c even if the shape of the Venus+ cell is clearly visible before ablation the image has high background/noise around the cells. A good membrane reporter line normally gives a clearer picture.

2) Concerning pSmad2 staining as an indicator of Nodal activity. The authors didn't get the antibody to work in pre-implantation embryos and they claimed the transgene A7-Venus is sufficient for measuring Nodal activity. This part is accepted. However, the Smad2-Venus image in Fig. 2a doesn't show a clear and specific signal. I would suggest removing it because even if the signal is specific, it doesn't indicate the activity of Activin/Nodal signalling.

3) In Fig.4 the quantification of cell number (shown as % of DAPI cells) is accepted but it would be more informative to show the real numbers (graphically with cell numbers for GATA6, OCT4,

VENUS) in supplementary Figures.

4) Concerning Suppl-Fig.5a, the authors state: "Cer11 seems to have a redundant role in AVE formation since Hex⁺ cells (AVE) were expanded in a portion of Lefty1,2^{-/-}, Cer11^{-/-} embryos examined (1/2 embryos) (Supplementary Fig. 5e). It may be that more VE cells were recruited to become AVE cells due to increased migration of DVE cells". Here, the number of analysed embryos is not clear. In addition, the last sentence is too speculative and is not supported by any data. I would suggest removing it otherwise new experiments using live imaging of AVE reporter lines would be required. Even the expansion of Hex cells (mentioned above) is not supported by any measurement.

5) The text contains several misspelling errors.

Major points:

There are still several major points the authors should address before the manuscript could be considered for publication. The following comments are aimed to suggest to the authors some experiments to unequivocally reinforce the main messages of the work that: a) Lefty expression in Epiblast (Epi) or primitive endoderm (PrE) is Nodal dependent; b) The selection of DVE cells in pre-implantation embryos is random.

1) In Fig. 2a, the authors show the expression of Cripto using the transgene Cripto/Tomato observing expression in the ICM at E3.5 and in the Epiblast at E4.5. I strongly suggest to perform immunostaining for Cripto in order to unequivocally confirm its expression is mandatory for proper Nodal signalling activation. Indeed, in Fiorenzano et al Nat Comm 2016, Cripto expression (RNA and protein) was described in Nanog-expressing cells at E3.5 and continuing in the Epi at E4.5. This point must be considered very important because the data presented by the authors suggests Lefty expression is Nodal-dependent in Epi and PrE cells. So it becomes crucial to show at the cellular level whether Nodal signalling components (i.e. Cripto), Nodal activity indicator (pSMAD2 or A7-Venus) and Lefty (endogenous protein) are co-expressed in PrE.

2) Concerning the use of antibody to show Lefty expression. The authors claim the antibody doesn't work. Nevertheless, clear staining is shown in the datasheet of the antibody and by Hoshino et al (Developmental Biology 2015) suggesting it is possible to use it for detection of endogenous staining.

3) Concerning the major request about following the fate of Lefty⁺ cells through the peri-implantation period, the authors unsuccessfully tried to use the IVC protocol described in Morris 2012 or Bedzhov 2014. It's not clear whether the authors have followed the method published in Nature Protocols by Bedzhov et al. that several other groups are now using successfully and where the efficiency of the in vitro development of embryos is shown. This point remains crucial to unequivocally confirm that the DVE-prospective cells (Lefty⁺ PrE cells) selected during pre-implantation are the source of the E5.5 DVE cells.

Even if the authors have successfully answered most of the points, it remains unclear if PrE cells can respond to Nodal and then express Lefty in a Nodal-dependent manner. The presented data (mostly obtained by transgenic lines) are somehow in contradiction with published data in particular where the expression of Cripto was not shown in PrE cells. This should be important for the proper activation of Nodal signalling. If Cripto is really demonstrated not to be expressed in PrE cells, then it remains unclear how expression of Lefty could be Nodal-dependent. Moreover, the authors did not attempt to follow the fate of L1-PE⁺ cells through peri-implantation until the DVE is specified.

In this reviewer's opinion the authors haven't provided strong data to support their major claims and the manuscript should be considered for publication only if further evidence is provided.

Reviewer #3:

Remarks to the Author:

I would recommend this paper for publication. However, there are some obvious spelling and style mistakes that need to be addressed and corrected prior to publication. Examples of the mistakes that

are to be fixed are:

- 1) Awkward phrasing/sentences in lines 44-48.
- 2) Grammatical error, line 53 (either "a maternal cue" or "maternal cues")
- 3) In numerous places authors refer to GATA6 as "GAT6" in the text. Please correct.

It is my recommendation that the editor check the manuscript thoroughly to catch other inconsistencies and typing mistakes prior to submission.

Reviewer #4:

Remarks to the Author:

I had a number of issues with the quality of the data in the original version of the manuscript and with the interpretation of some of the data. In the revised version the authors have adequately addressed all of my comments and criticisms and I think that the quality of the paper is definitely improved.

Response to the reviewer's comments

Reviewer #1 (Remarks to the Author):

Takaoka et al. have substantially improved their manuscript, and have addressed most of the reviewer's comments to satisfaction. However, some minor points still need to be addressed before the manuscript can be considered for publication.

Figure 3:

*Authors state in text that *lefty1* expression began either in the blastomere that received Nodal mRNA (6/19 embryos) or in a neighboring cell (12/19 embryos).*

*6+12=19 (100%). In Fig. 3g, however, authors state that 5% of embryos show *Lefty1* expression in more distant cells. Numbers here do not add up.*

Response: We forgot to mention the remaining one embryo (1/16 embryos: ~5%) that showed *Lefty1* expression in a distant cell. We now mention this in the text (page 7).

*Fig. 3d in this reviewer's mind shows rather the expression of *Lefty1(mVenus)* in a distant cell than as the authors claim in a neighboring cell. Also, the DAPI channel in this figure is overexposed so it is hard to make out single cells.*

Response: We have performed additional experiments and now provide an additional embryo showing *Lefty1(mVenus)* expression in a distant cell (Fig. 3d).

Reviewer #2 (Remarks to the Author):

In the revised manuscript Takaoka and Hamada have tried to address most of the criticisms raised by the referee but some points remain not explored and not well explained. The authors should address both minor and major points that haven't been addressed.

Minor points and comments:

The following points are considered minor and they are requested to improve the quality of the messages of the manuscript

1) Quality of the pictures: the authors have successfully improved the quality of the pictures nevertheless some further improvement would be required to clearly present the data. In general, the following panels seem to have strong artificial enhancement of the signal (very high contrast and brightness) and lack of suitable cellular resolution (that should be easily achieved using a confocal microscope as mentioned in the Methods in addition to presenting the DAPI signal to reveal the nucleus where not shown). Fig. 2a,c; Fig.3h (this image is not clear – Nodal protein and Cherry appear merged and the cellular resolution is of poor quality).

Response: We have improved Fig. 2a, 2c and 3h: merged views are shown for *Cripto(Tomato)* and *Smad2(Venus)*(Fig. 2a); replaced by a new embryo for Fig.2c; magnified views are shown for Fig.3h.

Fig. 7b strong contrast seems to have been used to hide the background/noise relative to the staining for Lefty and Cer11.

Response: The “background-looking” signals in Fig. 7b are not background/noise but are true signals that represent Cer11 expression in the definitive endoderm and Lefty1/2 expression in ectoderm cells. We have added arrowheads (magenta: definitive endoderm; green: definitive ectoderm) in Fig. 7b and mentioned in the legend.

Fig. 6c even if the shape of the Venus+ cell is clearly visible before ablation the image has high background/noise around the cells. A good membrane reporter line normally gives a clearer picture.

Response: We have improved Fig.6c, in response to the suggestion. Membrane is now positive for Venus before ablation.

2) Concerning pSmad2 staining as an indicator of Nodal activity. The authors didn't get the antibody to work in pre-implantation embryos and they claimed the transgene A7-Venus is sufficient for measuring Nodal activity. This part is accepted. However, the Smad2-Venus image in Fig. 2a doesn't show a clear and specific signal. I would suggest removing it because even if the signal is specific, it doesn't indicate the activity of Activin/Nodal signalling.

Response: We now provide a merged view for *Smad2* expression in Fig. 2a. *Smad2* expression is ubiquitous and does not show specific pattern. We would not insist, but would like to keep this image because we believe that this is an important point.

3) In Fig.4 the quantification of cell number (shown as % of DAPI cells) is accepted but it would be more informative to show the real numbers (graphically with cell numbers for *GATA6*, *OCT4*, *VENUS*) in supplementary Figures.

Response: We have added the real numbers in Supplementary Figure 6. Also we now provide the real number of DVE cells, PrE cells and total cells in Nodal mRNA-injection experiments shown in Fig. 3f (the real numbers are summarized in Supplementary Fig. 3b, c).

4) Concerning Suppl-Fig.5a, the authors state: “*Cer11* seems to have a redundant role in AVE formation since *Hex*⁺ cells (AVE) were expanded in a portion of *Lefty1,2*^{-/-}, *Cer11*^{-/-} embryos examined (1/2 embryos) (Supplementary Fig. 5e). It may be that more VE cells were recruited to become AVE cells due to increased migration of DVE cells”. Here, the number of analysed embryos is not clear. In addition, the last sentence is too speculative and is not supported by any data. I would suggest removing it otherwise new experiments using live imaging of AVE reporter lines would be required. Even the expansion of *Hex* cells (mentioned above) is not supported by any measurement.

Response: As suggested by this reviewer, we have removed data with *Lefty1,2*^{-/-}, *Cer11*^{-/-} embryos (previously, Supplementary Fig. 5e) and the corresponding sentence. Although not requested, we have increased the number of *Lefty1,2*^{-/-} embryos examined in Fig. 4a, b, d, e (summarized in Supplementary Fig. 6).

Major points:

There are still several major points the authors should address before the manuscript could be considered for publication. The following comments are aimed to suggest to the authors some experiments to unequivocally reinforce the main messages of the work that: a) Lefty expression in Epiblast (Epi) or primitive endoderm (PrE) is Nodal dependent; b) The selection of DVE cells in pre-implantation embryos is random.

1) In Fig. 2a, the authors show the expression of *Cripto* using the transgene *Cripto/ Tomato* observing expression in the ICM at E3.5 and in the Epiblast at E4.5. I strongly suggest to

perform immunostaining for Cripto in order to unequivocally confirm its expression is mandatory for proper Nodal signalling activation. Indeed, in Fiorenzano et al Nat Comm 2016, Cripto expression (RNA and protein) was described in Nanog-expressing cells at E3.5 and continuing in the Epi at E4.5. This point must be considered very important because the data presented by the authors suggests Lefty expression is Nodal-dependent in Epi and PrE cells. So it becomes crucial to show at the cellular level whether Nodal signalling components (i.e. Cripto), Nodal activity indicator (pSMAD2 or A7-Venus) and Lefty (endogenous protein) are co-expressed in PrE.

Response: Two lines of evidence indicate that PrE can receive Nodal signaling even though *Cripto* is not expressed in PrE.

1) *Cryptic*, a Nodal co-receptor functionally-redundant with *Cripto*, is expressed in PrE, while *Cripto* is expressed in the epiblast (Chu and Shen, 2010 Dev. Biol.).

2) *Cripto* and *Cryptic* are known to act non-cell autonomously, which has been shown at multiple stages of mouse embryos including the peri-implantation stage: i) Chu, J. *et al.* (2005). Non-cell-autonomous role for *Cripto* in axial midline formation during vertebrate embryogenesis. *Development* **132**, 5539–5551. ii) Chu and Shen (2010), Functional redundancy of EGF-CFC genes in epiblast and extraembryonic patterning during early mouse embryogenesis. *Dev. Biol.* 342:63-73. iii) Lee GH, Fujita M, Takaoka K, Murakami Y, Fujihara Y, Kanzawa N, Murakami KI, Kajikawa E, Takada Y, Saito K, Ikawa M, Hamada H, Maeda Y, Kinoshita T. (2016). A GPI processing phospholipase A2, PGAP6, modulates Nodal signaling in embryos by shedding CRIPTO. *J Cell Biol.* 215(5):705-718. The Lee et al (2016) paper shows that *Cripto* needs to be cleaved off into a soluble form by a GPI processing enzyme in early mouse embryo, which is the reason why *Cripto* (and *Cryptic*) act cell non-autonomously.

This is now mentioned on page 6~7.

2) *Concerning the use of antibody to show Lefty expression. The authors claim the antibody doesn't work. Nevertheless, clear staining is shown in the datasheet of the antibody and by Hoshino et al (Developmental Biology 2015) suggesting it is possible to use it for detection of endogenous staining.*

Response: We have used the same antibody. We were able to detect endogenous Lefty proteins at stages later than E5.5 (Fig.6a, d; Fig.7b; Supplementary Fig. 1c,d; Supplementary Fig.5b), but were unable to do so at earlier stages. However, we did detect Lefty1 expression with a variety of different BAC and plasmid transgenes (Venus, mVenus, Cherry, tdTomato, LacZ transgenes; multiple transgenic lines for each transgene). Most likely, *Lefty1* expression at the peri-implantation stage is weaker than that at later stages.

3) Concerning the major request about following the fate of Lefty+ cells through the peri-implantation period, the authors unsuccessfully tried to use the IVC protocol described in Morris 2012 or Bedzhov 2014. It's not clear whether the authors have followed the method published in Nature Protocols by Bedzhov et al. that several other groups are now using successfully and where the efficiency of the in vitro development of embryos is shown. This point remains crucial to unequivocally confirm that the DVE-prospective cells (Lefty+ve PrE cells) selected during pre-implantation are the source of the E5.5 DVE cells. Even if the authors have successfully answered most of the points, it remains unclear if PrE cells can respond to Nodal and then express Lefty in a Nodal-dependent manner. The presented data (mostly obtained by transgenic lines) are somehow in contradiction with published data in particular where the expression of Cripto was not shown in PrE cells. This should be important for the proper activation of Nodal signalling. If Cripto is really demonstrated not to be expressed in PrE cells, then it remains unclear how expression of Lefty could be Nodal-dependent. Moreover, the authors did not attempt to follow the fate of LI-PE+ cells through peri-implantation until the DVE is specified.

Response: As we describe above, PrE cells can receive Nodal signaling even though Cripto expression is absent in PrE cells because Cryptic is expressed in PrE and because Cripto and Cryptic can act cell-non-autonomously. As for the in vitro culture system, we agree that live imaging with in vitro culture system would strengthen our conclusion, but we previously felt that such experiments are beyond the scope of this paper because we have already shown by genetic fate mapping that the DVE-prospective cells (Lefty1^{ave} PrE cells) selected during pre-implantation are indeed the source of the E5.5 DVE cells (Takaoka et al., *Nat. Cell Biol.*, 2011).

Nonetheless, for our future work, we have attempted to establish the in vitro culture system once more. Unfortunately, culture of E3.5 embryos was not successful (0/6 embryos

developed into a 3D structure). On the other hand, culture of E4.5 embryos was partially successful: 6/48 embryos developed to embryos with an egg-cylinder shape. We had followed the development of 2 of such 6 embryos by live imaging, and found that *Lefty1*⁺ PrE cells at E4.5 continued to express *Lefty1* and become DVE-like cells in both cases (please see the photos below; also a movie is submitted as Supplemental Video 5). Thus, in vitro culture was partially successful but the efficiency of normal development was still low (6/48 embryos), which prevents us performing laser-ablation experiments with in vitro cultured embryos. We would like to show these data only to reviewers, because our in vitro culture system is not perfect yet and needs to be improved for future work (if required, however, we would be happy show these data in Supplementary Information).

Legend to this figure: Mouse embryos harboring *Lefty1(mVenus)* transgene were recovered at E4.5, and were subjected to the in vitro culture as described by Morris et al (2012) and Bedzhvov et al (2014). Embryos were followed by live imaging. Venus⁺ cells in PrE cells at E4.5 are found in the distal visceral endoderm at 26h after the culture. Some Venus⁺ cells have migrated toward the proximal side at 33.5 h.

Reviewers' Comments:

Reviewer #1:

Remarks to the Author:

Takaoka et al. have put in a lot of effort to address all reviewers' comments and have substantially improved their manuscript. This reviewer is satisfied with the revision and has no further suggestions for improvement. The manuscript now warrants publication.

Reviewer #2:

Remarks to the Author:

This manuscript still has serious flaws and its conclusion is not supported by the data presented.

For example, the title. The authors try to track cells that contribute to DVE. This might have some impact on specification of the Anterior but not directly on the Posterior of the embryo and yet their conclusions are not just about AVE specification but about the whole AP axis. This is potentially flawed.

Another example is the last sentence of their abstract, which shows how little this work contributes. It has already been known that the DVE does not come from a pre-pattern fixed before the blastocyst stage. Thus the authors offer us exactly the same conclusion to one that we already know. They should direct their conclusion to what exactly is their novel discovery here.

The authors referred to cells in the blastocyst stage embryo as blastomeres. This is confusing as the term blastomeres is normally restricted to large cells of the early cleavage stage embryo, not the inner cell mass cells of the blastocyst.

I think it is important that the authors correct these over-interpretations and inaccuracies before this manuscript is published in any journal

Further comments from Referee #4 - commenting on the points raised by Referee #2:

In terms of the previous comments of Referee 2, the authors have mostly satisfactorily answered the issues raised. I give details below.

For the minor points, the points are all answered except that the authors should show a merge of the Cripto and Smad2 in Figure 2a and in Figure 2C, lefty1 is not visible in the merge. In Figure 6c, in the 8h panel, Cer11 needs to be labelled.

For the major points, the main issue that the reviewer has is that he/she is not convinced that the lefty1 expression in the PrE is really Nodal dependent. He/she is concerned that Cripto, the Nodal co-receptor is not expressed there. However, the authors state correctly that the other EGF-CFC family member, Cryptic is expressed there and also that the EGF-CFCs can act cell non-autonomously. Taking the fact that the PrE cells express Cryptic with the data in Supplementary Figure 2 that show clearly that both the Epi and PrE expression of lefty1 is inhibited by the Nodal receptor inhibitor SB-431542, I am convinced that both expression domains of lefty1 are Nodal-dependent. The one thing the authors could do to strengthen this further would be to look at the early lefty1 expression in Cripto $-/-$, Cryptic $-/-$ and the double mutant. I don't know how feasible this would be and have to say that I don't think that this is crucial for this paper.

The referee also wants proof that the lefty1+ PrE cells become DVE. As the authors correctly point out, they have already shown this in a previous paper. Therefore, I do not see this as an issue. In addition, the in vitro culture experiment that the authors present looks very convincing, although I understand that as they only have very low numbers they do not want to publish these data until the system is more robust.

In my view, the authors can say something about the AP axis, as well as the DVE, as for their ablation experiments, where a new lefty1+ cell arises, they do look at both anterior and posterior

markers and show that the AP axis is normal.

In terms of the last sentence of the abstract that the referee disagrees with, I think it does sum up the paper, although the authors should make clearer both here and in the discussion exactly what they mean by selection of the DVE cells being both random and regulated, as it does sound contradictory.

In conclusion, I am still of the opinion that the work is well done and that, apart from the small issues mentioned above and the possible inclusion of *lefty1* expression in *Cripto/Cryptic* mutants that the paper is acceptable for publication.

Response to the reviewers' comments

Reviewer #1 (Remarks to the Author):

Takaoka et al. have put in a lot of effort to address all reviewers' comments and have substantially improved their manuscript. This reviewer is satisfied with the revision and has no further suggestions for improvement. The manuscript now warrants publication.

Our response: Thank you.

--

Reviewer #2 (Remarks to the Author):

This manuscript still has serious flaws and its conclusion is not supported by the data presented.

For example, the title. The authors try to track cells that contribute to DVE. This might have some impact on specification of the Anterior but not directly on the Posterior of the embryo and yet their conclusions are not just about AVE specification but about the whole AP axis. This is potentially flawed.

Another example is the last sentence of their abstract, which shows how little this work contributes. It has already been known that the DVE does not come from a pre-pattern fixed before the blastocyst stage. Thus the authors offer us exactly the same conclusion to one that we already know. They should direct their conclusion to what exactly is their novel discovery here.

The authors referred to cells in the blastocyst stage embryo as blastomeres. This is confusing as the term blastomeres is normally restricted to large cells of the early cleavage stage embryo, not the inner cell mass cells of the blastocyst.

I think it is important that the authors correct these over-interpretations and inaccuracies before this manuscript is published in any journal

Our response: We thank thoughtful suggestions. The title has been changed according to the suggestion: the word anterior-posterior is removed. The word “blastomere” has been changed to “cell” throughout the text.

Referee #4

1) *Fig. 2a: a merge of the Cripto and Smad2*

Our response: Smad2 is positive in all cells at this stage, so a merge of the Cripto and Smad2 would not provide any additional information. Therefore, we feel that a merged photo is not necessary.

2) *Fig. 2c: Lefty1 is not visible in a merged picture.*

Our response: We agree with the reviewer. We have improved the merged photo. Now, Lefty1 is visible in a merged photo.

3) *Fig. 6c: Cer11 needs to be labelled on the 8h panel.*

Our response: We have added correct labels in the 8h panels of Fig. 6c. Actually, the purple/pink staining in the second panel from the top of the 8h column was Gata6 (not Cer11).

4) Lefty1 expression in Cripto(-/-),Cryptic(-/-) mutant embryos.

Our response: Unfortunately, Cripto and Cryptic mutant mice are not currently available in my lab. However, it has been shown by others (Chu and Shen, Dev Biol. 2010) that Lefty1 expression at E5.5 (DVE cells) is/are indeed lost in Cripto(-/-),Cryptic (-/-) double mutant embryos. We will cite this paper in the text.